# Learning an Actionable Discrete Diffusion Policy via Large-Scale Actionless Video Pre-Training

Haoran He [1]    Chenjia Bai [2,4†]    Ling Pan [1]    Weinan Zhang [3]    Bin Zhao [4]    Xuelong Li [2,4]

[1] Hong Kong University of Science and Technology
[2] Institute of Artificial Intelligence (TeleAI), China Telecom
[3] Shanghai Jiao Tong University    [4] Shanghai Artificial Intelligence Laboratory

## Abstract

Learning a generalist embodied agent capable of completing multiple tasks poses challenges, primarily stemming from the scarcity of action-labeled robotic datasets. In contrast, a vast amount of human videos exist, capturing intricate tasks and interactions with the physical world. Promising prospects arise for utilizing actionless human videos for pre-training and transferring the knowledge to facilitate robot policy learning through limited robot demonstrations. However, it remains a challenge due to the domain gap between humans and robots. Moreover, it is difficult to extract useful information representing the dynamic world from human videos, because of its noisy and multimodal data structure. In this paper, we introduce a novel framework to tackle these challenges, which leverages a unified discrete diffusion to combine generative pre-training on human videos and policy fine-tuning on a small number of action-labeled robot videos. We start by compressing both human and robot videos into unified video tokens. In the pre-training stage, we employ a discrete diffusion model with a mask-and-replace diffusion strategy to predict future video tokens in the latent space. In the fine-tuning stage, we harness the imagined future videos to guide low-level action learning with a limited set of robot data. Experiments demonstrate that our method generates high-fidelity future videos for planning and enhances the fine-tuned policies compared to previous state-of-the-art approaches with superior performance. Our project webpage is available at `https://video-diff.github.io/`.

## 1   Introduction

How do we derive a general-purpose robot agent that can complete a wide variety of tasks? We believe that recent advances in vision and language give us a clue, which delves into pre-training foundation models on extremely large and diverse datasets, followed by fine-tuning on specific domains. For instance, through pre-training on internet-scale datasets [65], large language models [81, 82, 63] and vision models [9, 69, 72] showcase impressive performance on various downstream tasks such as question answering, coding, and image generation. However, unlike general visual language tasks that can exploit copious amounts of data available on the Internet, embodied tasks necessitate high-quality egocentric data in robotics domains for precise control. Collecting such data can be expensive or time-consuming due to the reliance on robot interactions through teleoperation or kinematic solvers [31], and significant gaps in embodiments and dynamics persist when applying them to different robots.

In contrast to the limited availability of robot data, there is a wealth of human interaction videos capturing intricate tasks and varied interactions with the physical world [27]. These videos inherently

---

[†]Correspondence to: Chenjia Bai (baicj@chinatelecom.cn).

38th Conference on Neural Information Processing Systems (NeurIPS 2024).

encapsulate rich semantic information regarding objects, environmental backgrounds, and hand-object interactions across diverse scenarios, making them potentially valuable for acquiring shareable knowledge relevant to embodied tasks.

Motivated by this, many works have emerged to learn various objectives pre-trained on human actionless videos, aiming to capture useful knowledge that can be beneficial for embodied tasks. These approaches involve learning pre-trained image representations [61, 57, 68, 91], trajectory representations [4, 85, 74], reward functions [13, 56] and world models [59, 89]. However, they are still limited to comprehending the dynamic rules of the world or reasoning based on long-term behavior rather than relying solely on step-by-step transitions. We summarize three main

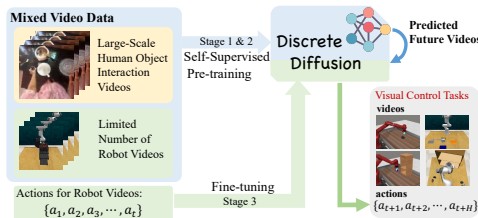

Figure 1: Overall framework of VPDD.

challenges that bottleneck their performance: (i) The domain gap between humans and robots which hinders knowledge transfer; (ii) Complex, diverse and noisy behavior patterns hidden in human videos which are difficult to learn; (iii) Large-scale data from different modalities (e.g., videos, actions, texts) which requires a scalable and high-expressive model architecture to process.

To address these challenges, we propose a **V**ideo-based **P**olicy learning framework via **D**iscrete **D**iffusion (VPDD). VPDD bridges the visual gap between the human and robot domains by representing these two data types as unified latent representations. Then, VPDD performs video prediction as a *pre-training stage* with actionless videos, which acquires the commonsense knowledge shared between human and robot interactions, including dynamic rules and behavior patterns (e.g., pick, place, push) to understand and complete tasks. Then, VPDD performs policy learning via a *fine-tuning stage* with action-labeled robot videos, where VPDD learns to predict actions with foresight from future video predictions. The pre-training stage learns extensive knowledge from human video prediction, and the fine-tuning stage concentrates on training parameters specifically associated with actions. To tackle the challenge of modeling the noisy and complex distribution of large-scale videos while enabling the multi-modal generation of both videos and actions, we leverage the generative capability and flexible architecture offered by discrete diffusion models [2, 29]. We provide an overview of our method in Fig. 1.

In summary, we highlight our contributions as follows. (i) We propose VPDD, a novel pretraining-finetuning paradigm for learning an actionable policy with limited robot data accessible. This paradigm demonstrates superior ability in transferring valuable knowledge from large-scale actionless human videos to downstream embodied tasks. (ii) We formulate both video prediction and action learning processes as unified discrete denoising problems, showing the supreme effectiveness in handling high-dimensional, multi-modal data. (iii) We conduct thorough experiments using human videos from Ego4D [27], as well as embodied datasets from Meta-World [98] and RLBench [43], showcasing its ability to predict dynamic-consistent future videos. Our actionable discrete diffusion policy also exhibits superior performance compared to previous state-of-the-art approaches [32, 61, 24, 75, 15], encompassing both seen and unseen scenes for multi-task robotic problems.

## 2 Preliminaries

### 2.1 Multi-Task POMDP

In this work, we consider a generalist vision-based agent that is capable of addressing multi-task predicaments, where the landscape is characterized by the inherent challenge of acquiring different skills across tasks and partial observability when dealing with image inputs. Given a specific task $\mathcal{T} \sim p(\mathcal{T})$, we further approach the problem as a task-specified Partially Observable Markov Decision Process (POMDP), defined as $(\mathcal{S}^{\mathcal{T}}, \mathcal{O}, \mathcal{A}, \mathcal{P}^{\mathcal{T}}, \mathcal{R}^{\mathcal{T}}, \mu^{\mathcal{T}}, \gamma)$. Here, $\mathcal{O}$ is a shared observation space as we use image observations for all tasks. We also assume all tasks share the same action space with the same embodiment.

## 2.2 Vector Quantized Model

In order to unify the feature space of both human videos and robot videos, we leverage the Vector Quantized Variational Auto Encoder (VQ-VAE) [83] to compress high-dimensional data points into information-rich discretized latent codes. Given a high-dimensional video segment $\boldsymbol{x} \in \mathbb{R}^{T \times H \times W \times C}$, the encoder $E$ first converts it to the temporal-spatial features $\boldsymbol{z} = E(\boldsymbol{x}) = \{z_{m,i,l}\} \in \mathbb{R}^{t \times h \times w \times d}$, where $t \times h \times w$ represents the encoded sequence length and is much smaller than $T \times H \times W$. Then we transfer the continuous features into discrete space by performing a nearest neighbors lookup in a codebook of embeddings $\mathcal{Z} = \{e_j\}_{j=1}^{J} \in \mathbb{R}^{J \times d}$ to obtain the tokens

$$z_q = \text{Quantize}(z_{m,i,l}) := \arg\min_J \|z_{m,i,l} - e_j\|_2^2, \tag{1}$$

where the video tokens $\boldsymbol{z}_q \in \mathbb{R}^{t \times h \times w \times d}$ can be faithfully reconstructed via a decoder, i.e., $\hat{\boldsymbol{x}} = G(\boldsymbol{z}_q)$. The encoder $E$, decoder $G$, and codebook $\mathcal{Z}$ can be trained end-to-end via the following loss function $\mathcal{L} = \|\boldsymbol{x} - \hat{\boldsymbol{x}}\|_1 + \|\text{sg}[E(\boldsymbol{x})] - \boldsymbol{z}_q\|_2^2 + \beta\|\text{sg}[\boldsymbol{z}_q] - E(\boldsymbol{x})\|_2^2$, where sg denotes stop gradient.

## 2.3 Discrete Diffusion Model

The discrete diffusion model was first proposed to deal with discrete state space with transitions converging to a binomial distribution [76], and then extended to multinomial diffusion with more options for transition matrices [36, 2]. In this work, we utilize discrete diffusion with the absorbing state for sequence prediction of discrete tokens. Besides $J$ tokens from a codebook, an additional [MASK] token is introduced. We denote $\boldsymbol{x}_k$ as a one-hot vector identifying the token index. The forward process from $\boldsymbol{x}_{k-1}$ to $\boldsymbol{x}_k$ follows a Categorical distribution of $\boldsymbol{Q}_k \boldsymbol{x}_{k-1}$, as

$$q(\boldsymbol{x}_k|\boldsymbol{x}_{k-1}) = \text{Cat}(\boldsymbol{x}_k; p = \boldsymbol{Q}_k \boldsymbol{x}_{k-1}) = \boldsymbol{x}_k^T \boldsymbol{Q}_k \boldsymbol{x}_{k-1}, \tag{2}$$

where $[\boldsymbol{Q}_k]_{m,n} = q(\boldsymbol{x}_k = m|\boldsymbol{x}_{k-1} = n) \in \mathbb{R}^{(J+1)\times(J+1)}$ is the Markov transition matrix from $k-1$ to $k$, which is formulated as:

$$\boldsymbol{Q}_{k-1\to k} = \begin{pmatrix} \alpha_k + \beta_k & \beta_k & \beta_k & \cdots & 0 \\ \beta_k & \alpha_k + \beta_k & \beta_k & \cdots & 0 \\ \beta_k & \beta_k & \alpha_k + \beta_k & \cdots & 0 \\ \vdots & \vdots & \vdots & \ddots & \vdots \\ \gamma_k & \gamma_k & \gamma_k & \cdots & 1 \end{pmatrix}, \tag{3}$$

where $\alpha_k \in [0, 1]$ is the probability of retaining the token, and each ordinary token has a probability of $\gamma_k$ to be replaced by [MASK] token, leaving a chance of $\beta_k = (1 - \alpha_k - \gamma_k)/J$ to be diffused. Importantly, due to the property of the Markov chain, we can derive the probability of $\boldsymbol{x}_k$ at arbitrary timestep directly from $\boldsymbol{x}_0$ as

$$q(\boldsymbol{x}_k|\boldsymbol{x}_0) = \boldsymbol{x}_k^T \overline{\boldsymbol{Q}}_k \boldsymbol{x}_0, \text{with } \overline{\boldsymbol{Q}}_k = \boldsymbol{Q}_k \cdots \boldsymbol{Q}_1. \tag{4}$$

Besides, the posterior of this diffusion process is tractable as $q(\boldsymbol{x}_{k-1}|\boldsymbol{x}_k, \boldsymbol{x}_0) = \frac{q(\boldsymbol{x}_k|\boldsymbol{x}_{k-1}, \boldsymbol{x}_0)q(\boldsymbol{x}_{k-1}|\boldsymbol{x}_0)}{q(\boldsymbol{x}_k|\boldsymbol{x}_0)} = \frac{(\boldsymbol{x}_k^T \boldsymbol{Q}_k \boldsymbol{x}_{k-1})(\boldsymbol{x}_{k-1}^T \overline{\boldsymbol{Q}}_{k-1} \boldsymbol{x}_0)}{\boldsymbol{x}_k^T \overline{\boldsymbol{Q}}_k \boldsymbol{x}_0}$. In the reverse process, rather than explic-itly predicting the posterior through a denoising neural network, the $\boldsymbol{x}_0$-parameterisation enhances stability and allows for fast inference (by skipping $\Delta k$ steps per iteration). The reverse transition with reparameterisation is formulated as

$$p_\theta(\boldsymbol{x}_{k-1}|\boldsymbol{x}_k) = \sum_{\tilde{\boldsymbol{x}}_0} q(\boldsymbol{x}_{k-1}|\boldsymbol{x}_k, \tilde{\boldsymbol{x}}_0)p_\theta(\tilde{\boldsymbol{x}}_0|\boldsymbol{x}_k), \tag{5}$$

where the neural network predicts the logits of the target data $q(\boldsymbol{x}_0)$.

# 3 Methodology

We commence with the pre-training of our model through future video prediction, enabling the learning of a general dynamic pattern across diverse domains. Subsequently, we fine-tune the model using a limited dataset of robot data for policy learning, leveraging foresight from predicted videos. Our framework is illustrated in Figure 2.

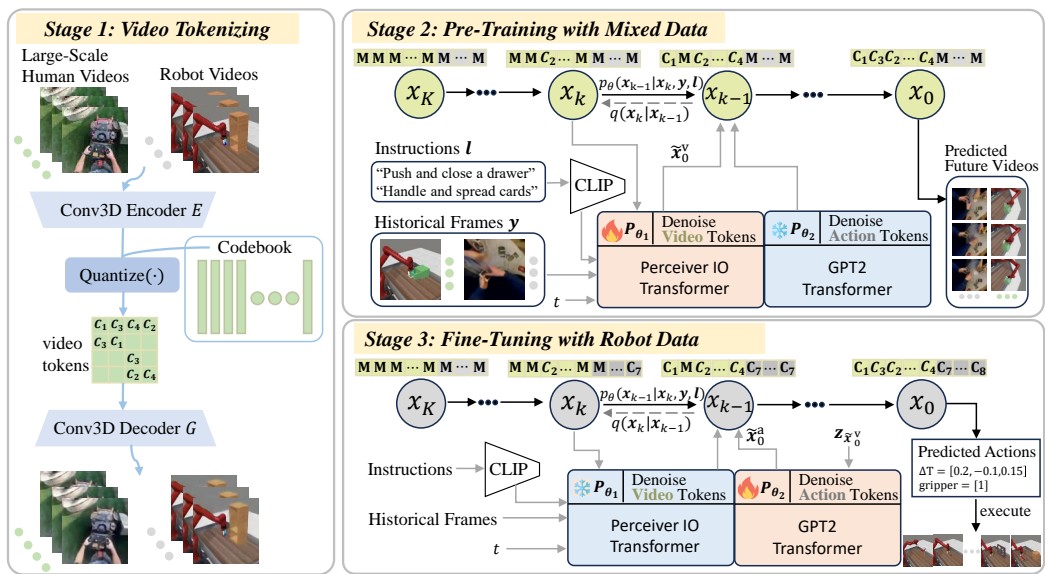

Figure 2: **Overall pipeline of VPDD.** A video-based VQ-VAE is leveraged to encode both human and robot videos into discrete latent codes. Subsequently, a unified discrete diffusion is firstly pre-trained on these video latent codes via a self-supervised objective, predicting future videos conditioning on language instructions and historical videos. The pre-trained video prediction model $p_{\theta_1}$ can capture temporal dynamics and task-specific representations. Lastly, we fine-tune our diffusion model on a limited number of robot data. In each diffusion step of the fine-tuning stage, we leverage $p_{\theta_1}$ to provide hidden representations $z_{\tilde{x}_0^v}$ to benefit downstream action learning with video foresight. This integration of video prediction and action learning is achieved through our unified discrete diffusion.

## 3.1 Data Preparing and Tokenizing

**Robot Data Collection.** We use the rule-based script policy to rollout 20 expert demonstrations for each task in Meta-World [98]. We also run VPDD on 16 tasks from RLBench [43], a more challenging benchmark involving 6-Dof manipulation that necessitates multi-view images from a 3D scene to select actions. Following the multi-view manipulation frameworks [24, 75], we utilize the script motion-planner to collect 10 demonstrations for each task. Each demonstration from robot-data is formulated as $\tau_i = \{v_1, a_1, \cdots, v_t, a_t, \cdots, v_T, a_T\}$ with $v_t = [o_{t-I+1}, \cdots, o_{t-1}, o_t]$, where we set $I = 4$ throughout the paper and $a$ denotes the actions. In the context of Meta-World, $o_t$ represents a single-view RGB image at timestep $t$. For RLBench, $o_t = \{o_t^{\text{front}}, o_t^{\text{left}}, o_t^{\text{right}}, o_t^{\text{wrist}}\}$ comprises 4 RGB multi-view images (i.e., front, left shoulder, right shoulder, and wrist). Consequently, in RLBench, $v_t$ is formulated as $v_t = \{v_t^{\text{front}}, v_t^{\text{left}}, v_t^{\text{right}}, v_t^{\text{wrist}}\}$.

**Human Data Collection.** As for human data collection, we obtain untrimmed videos from the open-sourced Ego4D dataset [27], which contains massive-scale human-object interactions of various durations ranging from 5 seconds to 7 hours. We filter out videos without human-object interaction and segment each video into short clips with 8-frame intervals [60]. Thus each video is represented as $\tau_i = \{v_1, v_2, \cdots, v_n\}$, where $v_t$ denotes a clip containing 4 frames. This approach yields a total of 996,177 clips of human videos, comprising approximately 4M frames. More details on data collection and processing are given in §C.1.

**VQ-VAE Encoding.** To extract useful features from raw videos in both human and robot domains, a conventional approach is to directly encode them into an embedding space using pre-trained vision models like ViT. However, these models are usually specifically trained on image dataset [71], posing a significant challenge due to the domain gap with our interaction videos. Thus, we leverage VQ-VAE to compress the diverse and noisy videos into discrete latent codes, which provide a unified codebook for mixed videos and alleviate the domain gaps between human and robot videos. Formally, we adopt the VQ-VAE architecture introduced by VideoGPT [94] for encoding videos into a discrete latent space. The codebook comprises 2048 codes, each represented by 256-dimensional embeddings. The encoder architecture consists of a series of 3D convolutions that downsample by a factor of 4 over space-time (resulting in a $64\times$ total reduction), followed by 6 attention residual blocks. Consequently,

each video clip $v_t \in \{\tau_i\}$ is embedded into latent codes $e_t$. The architecture for the decoder is the reverse of the encoder, featuring attention residual blocks followed by an equivalent number of 3D transposed convolutions for upsampling over space-time. The VQ-VAE is pre-trained on large-scale videos and remains fixed in the subsequent processes, providing flexibility for various downstream utilization methods.

**Action Discretizing.** For subsequent pre-training and fine-tuning, we process the collected continuous actions via uniform action discretization [46, 11]. In the case of Meta-World, the action space is a 2-tuple consisting of the change in the 3D space of the end-effector followed by a normalized torque that the gripper fingers should apply. Here all the continuous dimensions are discretized into 48 bins uniformly. Thus, the robot action can be represented using ordinals of the discrete bins as a 4 integer number. For RLBench, an action consists of the gripper open state and 6-DoF pose including position and rotation. The position is discretized into 360 bins, and rotation is discretized into Euler angles as 1-degree bins for each of the 3 rotation axes [75]. Gripper open state is a binary value.

### 3.2 Video Prediction via Unified Discrete Diffusion

Extracting general patterns useful for downstream decision-making from large-scale in-the-wild human videos is challenging, primarily because of the absence of labeled actions and the complexity of the underlying structure of human interactions. Different from previous ways of learning a visual representation, we propose a novel objective to further unleash the representation and temporal modeling ability of diffusion models. Specifically, after obtaining discrete tokens from VQ-VAE encoding, we train a unified discrete diffusion model on the latent space via a self-supervised objective. This objective involves predicting future videos based on observed historical videos for both humans and robots, while masking action tokens. Benefiting from the proposed objective and the $x_0$-parameterisation of discrete diffusion, the diffusion model is incentivized to capture both the high-level temporal dynamics and the low-level visual commonalities between historical and future videos at each diffusion step. Then the acquired knowledge can be leveraged to guide action denoising at each step.

**Unified Transition Matrix.** The presence of a transition matrix determines the nature of the discrete diffusion model [2]. While the original discrete diffusion is limited to one modality, drawing inspiration from UniD3 [37], which enhances the transition matrix to encompass both images and text, we construct a unified transition matrix to capture global connections between the two modalities—videos and actions. The matrix $[\boldsymbol{Q}_k]_{m,n}$ below illustrates the unified transition process:

$$\boldsymbol{Q}_k = \begin{bmatrix} \alpha_k + \beta_k & \beta_k & \cdots & \beta_k & 0 & 0 & \cdots & 0 & 0 \\ \beta_k & \alpha_k + \beta_k & \cdots & \beta_k & 0 & 0 & \cdots & 0 & 0 \\ \vdots & \vdots & \ddots & \vdots & \vdots & \vdots & \ddots & \vdots & \vdots \\ \beta_k & \beta_k & \cdots & \alpha_k + \beta_k & 0 & 0 & \cdots & 0 & 0 \\ 0 & 0 & \cdots & 0 & \alpha_k + \beta_k^* & \beta_k^* & \cdots & \beta_k^* & 0 \\ 0 & 0 & \cdots & 0 & \beta_k^* & \alpha_k + \beta_k^* & \cdots & \beta_k^* & 0 \\ \vdots & \vdots & \ddots & \vdots & \vdots & \vdots & \ddots & \vdots & \vdots \\ 0 & 0 & \cdots & 0 & \beta_k^* & \beta_k^* & \cdots & \alpha_k + \beta_k^* & 0 \\ \gamma_k & \gamma_k & \cdots & \gamma_k & \gamma_k & \gamma_k & \cdots & \gamma_k & 1 \end{bmatrix},$$

where $\beta_k$ and $\beta_k^*$ are the probabilities of a token to be replaced by any other accessible tokens in different modalities. The dimension of $\boldsymbol{Q}_k$ is $(J + J^* + 1) \times (J + J^* + 1)$, where $J$ and $J^*$ are the number of tokens in different modalities, i.e., $J$ is the size of codebook in VQ-VAE and $J^*$ is the number of action classes in discretization. The sum of each column in this transition matrix is one to preserve probability mass. Mathematically, we have $\beta_k = (1 - \alpha_k - \gamma_k)/J$ and $\beta_k^* = (1 - \alpha_k - \gamma_k)/J^*$. All the mass of the stationary distribution falls on the [MASK] token, which satisfies the prerequisite for a discrete diffusion model transition matrix [2]. The details of the diffusion process are provided in §A.1.

**Unified Objective.** We cast both video prediction and action learning as a conditional generative problem, and the goal is to maximize $\mathbb{E}_{\tau \sim \cup_i \tau_i}\big[\log p_\theta(\boldsymbol{x}_0(\tau) \mid \boldsymbol{y}(\tau), \boldsymbol{l})\big]$. Here $\boldsymbol{x} = [\boldsymbol{x}^{\mathrm{v}}, \boldsymbol{x}^{\mathrm{a}}]$, where $\boldsymbol{x}^{\mathrm{v}} = [\boldsymbol{e}_{t+h+1}, \cdots, \boldsymbol{e}_{t+h+M}]$ represents future video segments with $M$ clips, and $\boldsymbol{x}^{\mathrm{a}} = [a_t, \cdots, a_{t+H-1}]$ denotes action sequences of $H$ steps. $\boldsymbol{y} = \boldsymbol{e}_t$ serves as the condition containing historical video tokens. $\boldsymbol{l}$ is the language instructions describing current tasks. In practice, we train two separate denoising networks, namely $p_{\theta_1}(\boldsymbol{x}_{k-1}^{\mathrm{v}}|\boldsymbol{x}_k, \boldsymbol{y}, \boldsymbol{l})$ and $p_{\theta_2}(\boldsymbol{x}_{k-1}^{\mathrm{a}}|\boldsymbol{x}_k, z_{\tilde{\boldsymbol{x}}_0^{\mathrm{v}}}, \boldsymbol{l})$, to learn videos and actions, respectively. Here, $z_{\tilde{\boldsymbol{x}}_0^{\mathrm{v}}}$ represents the hidden representation of predicted future videos given by $p_{\theta_1}$ at each diffusion step, which is utilized to guide action learning. Formally, $\tilde{\boldsymbol{x}}_0^{\mathrm{v}} = \mathrm{Softmax}(\mathrm{MLP}(z_{\tilde{\boldsymbol{x}}_0^{\mathrm{v}}}))$.

As the actions are absent during the pre-training stage, we mask $\boldsymbol{x}^{\mathrm{a}}$ and freeze $p_{\theta_2}$, as illustrated in Figure 2. The network is trained to minimize the following variational lower bound (VLB) [76, 29]:

$$\mathcal{L}_{\mathrm{vlb}} = \mathcal{L}_0 + \sum\nolimits_{k=2}^{K} \mathcal{L}_{k-1} + \mathcal{L}_K, \tag{6}$$

where

$$\mathcal{L}_0 = -\mathbb{E}_{q(\boldsymbol{x}_1|\boldsymbol{x}_0)} \left[ \log p_{\theta_1}(\boldsymbol{x}_0^{\mathrm{v}}|\boldsymbol{x}_1, \boldsymbol{y}, \boldsymbol{l}) + \log p_{\theta_2}(\boldsymbol{x}_0^{\mathrm{a}}|\boldsymbol{x}_1, z_{\tilde{\boldsymbol{x}}_0^{\mathrm{v}}}, \boldsymbol{l}) \right], \tag{7}$$

$$\mathcal{L}_{k-1} = \mathbb{E}_{q(\boldsymbol{x}_k|\boldsymbol{x}_0)}[D_{\mathrm{KL}}(q(\boldsymbol{x}_{k-1}|\boldsymbol{x}_k, \boldsymbol{x}_0) \,\|\, [p_{\theta_1}(\boldsymbol{x}_{k-1}^{\mathrm{v}}|\boldsymbol{x}_k, \boldsymbol{y}, \boldsymbol{l}); p_{\theta_2}(\boldsymbol{x}_{k-1}^{\mathrm{a}}|\boldsymbol{x}_k, z_{\tilde{\boldsymbol{x}}_0^{\mathrm{v}}}, \boldsymbol{l})])], \tag{8}$$

$$\mathcal{L}_K = \mathbb{E}_{q(\boldsymbol{x}_0)} \left[ D_{\mathrm{KL}} \left( q(\boldsymbol{x}_K|\boldsymbol{x}_0) \,\|\, p(\boldsymbol{x}_K) \right) \right]. \tag{9}$$

$\mathcal{L}_K$ is a constant number that can be ignored in the training, as the prior distribution $p(\boldsymbol{x}_K)$ is fixed:

$$p(\boldsymbol{x}_K) = [\bar{\beta}_K, \bar{\beta}_K, \cdots, \bar{\beta}_K^*, \bar{\beta}_K^*, \cdots, \bar{\gamma}_K]. \tag{10}$$

**Model Architecture.** As in Eq. (8), the neural network $p_{\theta_1}$ receives $\boldsymbol{x}_k$, history $\boldsymbol{y}$, language $\boldsymbol{l}$ and the diffusion timestep $k$ as inputs. These inputs are individually embedded into embeddings $h$ of size $d$ via separate MLPs $f$, depicted as:

$$h_l = f_l(\mathrm{CLIP}(\boldsymbol{l})), \ h_{Ti} = f_{Ti}(k), h_{\boldsymbol{x}_k} = f_{\boldsymbol{x}_k}(\boldsymbol{x}_k), \ h_y = f_y(\boldsymbol{y}),$$

where language instructions $\boldsymbol{l}$ is encoded with CLIP's language encoder [66]. Afterwards, the embeddings are formulated as input tokens as $h_{\mathrm{tokens}} = \mathrm{LN}(h_{Ti} \times [h_l, h_{Ti}, h_{\boldsymbol{x}_k}, h_y] + E^{\mathrm{pos}})$, where $E^{\mathrm{pos}}$ is the positional embedding, and LN denotes layer normalization [3] for stabilizing training. The input sequence that represents a video can be extremely long so a standard Transformer with $\mathcal{O}(n^2)$ complexity is hard to fit. We adopt Perceiver Transformer [41] to tackle this problem, as it has been widely utilized for modeling long sequences [75, 24]. Perceiver is a latent-space Transformer, where instead of attending to the entire input, it computes cross-attention between the input and a much smaller set of latent vectors (which are randomly initialized and trained). These latents are encoded with self-attention layers, and are cross-attended with the input to match the size for the final outputs. More details about the Perceiver Transformer are referred to §A.2.

### 3.3 Learning to Act via Few-Shot Fine-Tuning

During the fine-tuning stage, we leverage a limited dataset of robot data, including both videos and actions, for rapid adaptation. Both $\boldsymbol{x}^v$ and $\boldsymbol{x}^a$ attend to training the diffusion model. Given that $p_{\theta_1}$ has been trained sufficiently to capture fruitful information to predict future videos from history, we freeze $p_{\theta_1}$ and solely tune parameters of $p_{\theta_2}$ to minimize $\mathcal{L}_{\mathrm{vlb}}$. As expressed in Eq. (8), the input of $p_{\theta_2}$ consists of $\boldsymbol{x}_k$, language $\boldsymbol{l}$, hidden representation $z_{\tilde{\boldsymbol{x}}_0^{\mathrm{v}}}$, and diffusion timestep $k$. In this case, we are tasked with predicting a sequence of action tokens $\boldsymbol{x}_0^{\mathrm{a}}$, considerably shorter than video-token sequence $\boldsymbol{x}_0^{\mathrm{v}}$, so we employ GPT2 [67] Transformer to process tokens embedded with MLPs. GPT2 has demonstrated an impressive ability to solve multi-task problems and model multimodal distributions. The model architecture of $p_{\theta_2}$ closely resembles that of MTDIFF-P [32]. More details of our method can be found in Appendix A.3.

## 4 Related Work

**Robot Learning from Human Videos.** Leveraging human videos [26, 23, 18, 27] for policy learning is promising to extract commonsense knowledge from human activities, which can be shared to embodied scenarios that suffer from scarce robot data [17, 22]. Since the human data is actionless and the domain gap between humans and robots exists, a main branch of research employs human video to learn shareable visual representations [12, 47] via time-contrastive [57], video-language alignment [61, 48], value function [8], and perceptual skills [40, 52]. Visual affordance like human-object interaction hotspots [54, 25] and the post-grasp trajectory [5] are also helpful for embodied agents in goal-conditioned imitation. Alternative methods involve extracting hand trajectories [4, 85] or keypoints [93] to transfer plans to robots. Different from the above methods, we eliminate the domain gaps by learning video tokens and representations for video prediction, which implicitly captures visual features, affordances, and long-term plans. Other works attempt to infer actions from videos via inverse kinematics [6, 50], whereas we learn action prediction through policy fine-tuning without external models.

**Pretraining for Generalized Policy Learning.** Early works of policy adaptation emerged in meta-RL [30, 99], while the pre-training and fine-tuning environments are assumed to be similar. Leveraging the transformer architecture, works perform pre-training in multi-task datasets by optimizing the multi-task policy [51, 70, 79, 80] or self-supervised objectives [78, 73, 59]. In tasks involving visual observations, methods adopt visual tokens for transformer-based multi-task policy learning [10, 11] and adaptation [53]. Additionally, some studies pre-train a reward function through video-language correspondence [13, 56] or diffusion models [21, 38] for downstream RL training. Different from the previous Transformer and continuous diffusion frameworks, our work first integrates visual tokens with discrete diffusion to predict consistent videos and actions simultaneously. Concurrently, GR-1 [88] utilizes human data to pre-train a GPT-style architecture for predicting future observations. In contrast, we perform video prediction instead of step-by-step image prediction using a unified discrete diffusion architecture.

**Diffusion Models for RL.** Diffusion models are a powerful family of generative models [72, 69] that can be categorized into continuous Gaussian diffusion models and discrete diffusion models that handle discrete visual tokens or symbols [29]. Continuous diffusion models have found extensive applications as multi-modal policies [87, 64, 16], environmental dynamics [95], and planners to generate action [45, 90] or state sequences [1, 14], guided by desired properties. Several methods also extend the diffusion models for multi-task learning [32, 62, 19] with low-dimensional states, while we specifically address the more challenging image-based setting. UniPi [20] and its following work [100] are related to our method by performing video generation via continuous diffusion, while we adopt a more flexible architecture with discrete diffusion to seamlessly connect the pre-training and fine-tuning stages, without relying on a task-specific inverse-dynamic model for acting. Additionally, we train the model on video tokens that maintain temporal consistency, while UniPi relies on super-resolution to improve the time consistency of generated frames.

## 5 Experiments

### 5.1 Bnechmarks and Baselines

After the video pertaining in Ego4D [27], we use the following robotic benchmarks to evaluate our method.

**Meta-World.** The Meta-World benchmark [98] contains 50 distinct manipulation tasks that require a Sawyer robot to interact with various objects with different shapes, joints, and connectivity. The action is the 3D position movements of the robot's end effector and the gripper openness. We follow recent works [96, 77] to extend the original environment to a more challenging setting with random goals, and refer to it as MT50-rand. We train the policy with 20 demonstrations per task, and report the average success rates on 50 evaluation episodes per task.

**RLBench.** RLBench [43] is a more challenging 3D manipulation benchmark with diverse tasks concerning interactions with a wide range of objects. We select 16 tasks from RLBench to evaluate our method, where each task has several possible variations, such as the shapes, colors, sizes and positions of objects. The input observations are captured from four RGB cameras positioned at the front, left shoulder, right shoulder, and wrist. The action is an 8-dimensional vector including 3-dimensional transitions, 4-dimensional quaternion, and a binary value about gripper openness. We follow the convention by using macro steps [42], which are key turning points in the action trajectory where the gripper changes its state (open/close) or the joint velocities approach to zero. We train the policy with 10 demonstrations per task and report average success rates on 25 evaluation episodes per task.

**Baselines for Meta-World.** We compare the proposed method VPDD with the following baselines: (i) **R3M-Diffusion** is a discrete diffusion model sharing identical architecture with $p_{\theta_2}$, leveraging the R3M [61] ResNet50 encoder to encode images as input. R3M is also trained on Ego4D videos via a contrastive learning objective and stands as the state-of-the-art (SOTA) visual representation specifically designed for manipulation tasks; (ii) **VC-1-Diffusion** utilizes VC-1 [58] encoder (ViT-L) to extract image representations, which is also trained on large-scale egocentric videos [27] and ImageNet [71] using Masked Auto-Encoding [33]. (iii) **MTDIFF-P** [32] is the SOTA method for multi-task RL, which employs a continuous diffusion model with a Transformer architecture. Since it is designed to handle state-based input, we employ the R3M encoder to extract hidden embeddings from images, which are then fed into MTDIFF-P; (iii) **Video-MTDT** is an extension of Decision Transformer (MT) [15], learning from multi-task data with video tokens as input; (v) **VPDD-w/o.-**

| Method | slide block | sweep to dustpan | meat off grill | turn tap | put in drawer | close jar | drag stick | stack blocks |
|---|---|---|---|---|---|---|---|---|
| PERACT (10 demos) [75] | 32 | 72 | 68 | 72 | 16 | 32 | 36 | 12 |
| RVT (10 demos) [24] | 54.67 ± 1.89 | **76.0 ± 3.27** | 69.33 ± 6.80 | **96.0 ± 3.27** | **91.33 ± 6.60** | 22.67 ± 4.99 | **97.33 ± 1.89** | 5.33 ± 1.89 |
| VPDD (Ours) | **70.67 ± 1.89** | 40 ± 3.27 | **73.33 ± 4.99** | 88.67 ± 3.77 | 24.0 ± 3.72 | **37.33 ± 8.22** | 66.67 ± 1.89 | **56.0 ± 5.66** |

| Method | screw bulb | put in safe | place wine | put in cupboard | push buttons | insert peg | stack cups | place cups |
|---|---|---|---|---|---|---|---|---|
| PERACT (10 demos) [75] | 28 | 16 | 20 | 0 | 56 | 4 | 0 | 0 |
| RVT (10 demos) [24] | 28.0 ± 3.27 | 38.67 ± 4.99 | 45.33 ± 6.80 | 5.33 ± 1.89 | **65.33 ± 1.89** | 2.67 ± 1.89 | 2.67 ± 1.89 | 0 ± 0 |
| VPDD (Ours) | **61.33 ± 6.80** | **70.67 ± 1.89** | **60.0 ± 6.53** | **30.67 ± 4.99** | 58.67 ± 1.89 | **73.33 ± 1.89** | **56.0 ± 5.66** | **30.67 ± 1.89** |

Table 1: Success rates (mean and std %) across 3 random seeds of various multi-task agents trained with 10 demonstrations and evaluated on 25 episodes per task. VPDD (Ours) outperforms SOTA methods of RLBench, i.e., PERACT and RVT, with an average improvement of $1.24\times$. Note that VPDD only takes RGB images as input while both RVT and PERACT utilize additional depth images as inputs.

**human** excludes human videos during pre-training, remaining other parts unchanged. This baseline helps to ablate the effect of pre-train on large-scale human videos; (vi) **SODA** [39] is a recently proposed diffusion-based representation method that employs an encoder to generate representations $z$ from input images to support the denoising process. We pre-train the encoder by employing the video prediction objective on the same dataset and subsequently feed the learned $z$ into $p_{\theta_2}$ during fine-tuning. See more details in §B.

**Baselines for RLBench.** Learning policies for RLBench is more challenging as it requires understanding the 3D scene structure for predicting the 6D poses of end-effectors. The baselines used in Meta-World all fail in the benchmark since they are disadvantaged with single-view observations. In contrast, VPDD can predict multi-view images, implicitly recovering the 3D geometry in manipulation. Thus, we use the following SOTA imitation architectures designed for 3D manipulation: (i) **RVT** [24] stands as the SOTA method, initially re-rendering visual observations into orthographic projections of cube views and subsequently predicting the next move based on these multi-view projections.; (ii) **PERACT** [75] encodes RGB-D images into voxel grid patches for 3D representation and predicts the action using the Perceiver Transformer.

## 5.2 Results Analysis

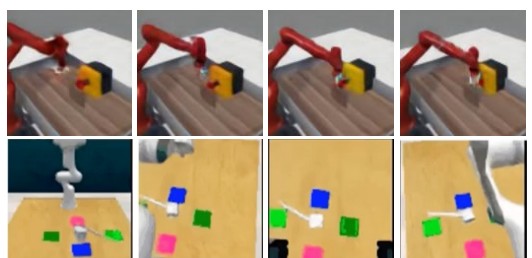

Figure 3: Single-view and multi-view images from Meta-World *button-press* and RLBench *drag-stick* tasks, sampled from videos predicted by $p_{\theta_1}$.

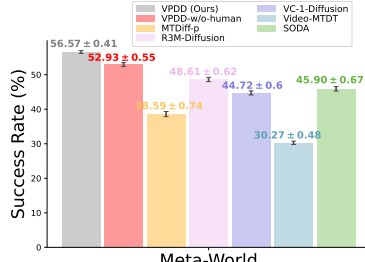

Figure 4: Average success rate across 3 seeds on MT50-rand. Each task is evaluated for 50 episodes.

**Question 1.** *Does our pre-trained diffusion model (i.e., $p_{\theta_1}$) capable of generating dynamic-consistent future videos?*

Although our primary contribution is not about video generation, it remains crucial to predict consistent future videos to aid in policy fine-tuning. The predicted raw videos are visually depicted in Fig. 3, with more video samples including real robots (trained by RoboSet data [7]) accessible at https://video-diff.github.io. These videos are reconstructed from predicted video tokens by using the decoder of VQ-VAE. After pre-training, the video-prediction model $p_{\theta_1}$ demonstrates the capability to generate dynamically consistent single-view videos for Meta-World and multi-view videos for RLBench. Furthermore, we computed the Frechet Video Distance (FVD) score [34, 86] averaged across frames on 32 generated video samples from the Meta-World dataset. From the results in Table 5.2,

| Method | FVD (↓) |
|---|---|
| UniPi [20] | 264.66 |
| VPDD (Ours) | **235.81** |

Table 2: Comparison of FVD score.

we observe that the FVD score of VPDD is considered acceptable and even lower than the hierarchical video synthesizing method UniPi [20] (score reported from its original paper). We attribute the

capability of generating high-quality videos to the well-trained video codebook and the proposed discrete diffusion model learned from large-scale human data.

**Question 2.** *How does VPDD compare to other offline baselines for vision-based multi-task decision-making?*

To evaluate the learned policy after fine-tuning, we take the first action generated by $p_{\theta_2}$ to interact with the environment. The results on Meta-World and RLBench are referred to Fig. 4 and Table 1 respectively, yielding the following key findings: (i) VPDD outperforms other SOTA methods in success rate by a large margin. For Meta-World, VPDD performs the best across 50 tasks with random goals. For RLBench, VPDD even outperforms the SOTA imitation architectures based on voxel and multi-view representations that are carefully designed for 3D manipulation, which usually require point clouds or 3D world rendering for scene understanding. Notably, VPDD achieves a remarkable success rate on *put in cupboard*, *insert peg*, *stack cups* and *place cups*, while both RVT and PERACT struggle on these challenging tasks. This verifies the efficacy of video pre-training for few-shot policy fine-tuning; (ii) According to Fig. 4, VPDD obtains a 6.9% relative improvement through pre-training on

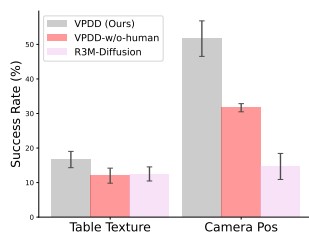

Figure 5: Average success rate across 3 seeds on shifted *button-press-v2* and *handle-press-v2* tasks.

both human and robot videos compared with VPDD-w/o.-human. Furthermore, VPDD surpasses R3M and VC-1 with a notable 16.4% and 26.5% higher success rate, demonstrating the potential of our diffusion representation via video prediction; (iii) R3M-Diffusion outperforms MTDIFF-P which employ R3M encoder with continuous diffusion architecture by 26.0%, and Video-MTDT with Transformer architecture by 60.6%. This highlights the superior capacity of discrete diffusion models compared to other model architectures.

**Question 3.** *How does VPDD generalize to unseen scenes?*

As suggested in recent works [92, 97], we evaluate the generalization ability of our model in the two most challenging settings, i.e., camera view and visual background. Specifically, we alter the camera position and table texture in the visual scene of Meta-World to assess the generalization ability of our method. According to Fig. 5, VPDD exhibits superior generalizability, attributed to the training of the diffusion representation on large-scale diverse human videos. Regarding the shift in camera position, VPDD outperforms the *VPDD-w/o.-human* by **63**% and R3M by **252**%. Moreover, we show that VPDD can generalize on different tasks with a competitive performance, demonstrating its potential to serve as a foundation model. Detailed results and settings can be found in §C.3.

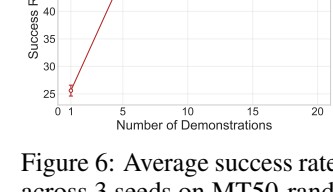

Figure 6: Average success rate across 3 seeds on MT50-rand, where VPDD is trained on a different number of demonstrations.

**Question 4.** *Can VPDD maintain satisfactory performance when provided with fewer robotic demonstrations?*

Leveraging the large-scale video pre-training, VPDD can learn the policy using only a small number of demonstrations. To validate the sample efficiency of our method, we conduct an ablation study on the number of demonstrations used in the fine-tuning stage. The results, depicted in Fig. 6, reveal that the performance of VPDD exhibits linear growth after training on 5 or more demonstrations, indicating the potential for VPDD to achieve better performance with increased demonstration data. Moreover, VPDD maintains a comparable success rate even when only **1** demonstration is used in the fine-tuning process.

**Question 5.** *How does VPDD perform when trained on different amounts of human data?*

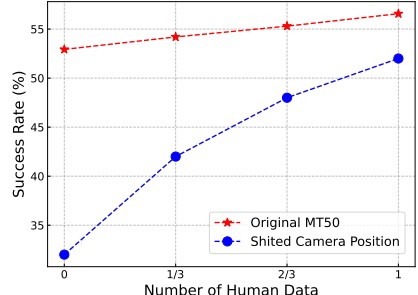

Figure 7: Ablation on the number of human videos during the pre-training stage, where the red curve is evaluated using the same experimental setting as in Fig. 4, and the blue curve corresponds to the setting in Fig. 5.

The superior generalizability of VPDD, which is validated in Fig. 5, stems from large-scale human-data pertaining, which effectively extracts commonsense

knowledge and representations that can be shared between human and unseen robot scenarios. To further investigate the effects of human-data pre-training, we ablate the number of human videos used during pre-training and present the performance changes in Fig. 7. Our findings indicate that an increased number of human videos enhances success rates, particularly in improving generalizability by a large margin.

**Question 6.** *Does VPDD outperforms other diffusion- based representation learning method?*

To verify the representation capability inherent in our unified discrete diffusion framework, we reproduce SODA [39], which serves as a strong diffusion representation method. As shown in Fig. 4, VPDD outperforms SODA in the context of policy fine-tuning. We hypothesize that VPDD provides coarse-to-fine representations (i.e., $z_{\tilde{x}_0^v}$) throughout the action-denoising steps from $K \rightarrow 1$, exactly encapsulating useful information the denoising network focuses on at each step [35]. In contrast, SODA produces representation $z$ that remains constant across all steps during action denoising.

## 6 Conclusion

We propose VPDD, a video-based policy learning framework via discrete diffusion. With a VQ-VAE tokenizer, we bridge the gap between human and robot videos by a discrete latent codebook. We leverage a unified discrete diffusion for pre-training on large-scale actionless mixture videos and subsequent fine-tuning the policy on a limited number of robot demonstrations. Experiments demonstrate that VPDD achieves superior performance on a variety of challenging manipulation tasks and showcases impressive generalization ability benefited from human video prediction. More Discussions of our work are given in §D.

## Acknowledgments

This work is supported by Shanghai Municipal Science and Technology Major Project (2021SHZDZX0102) and National Natural Science Foundation of China (62322603 & 62306242). We also thank the anonymous reviewers for their valuable suggestions.

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

## A  The Details of VPDD

### A.1  Discrete Diffusion

Discrete diffusion models with a mask-and-replace strategy generate sequences from [MASK] tokens. In this paper, they are trained by sampling a sequence $x_0 = [x^{\mathrm{v}}, x^{\mathrm{a}}]$, masking tokens according to a linear schedule [29] that corresponds to increasing the probability of being in the absorbing state linearly over time, and learning to predict the masked tokens given language $l$ and historical videos. Specifically, for the proposed unified transition matrix $Q_k$ in Eq. (6), the computation of diffusion process $q(x_k|x_0)$ in Eq. (4) can also be obtained in the following closed-form [37]:

$$\overline{Q}_k x_0 = \overline{\alpha}_k x_0 + \left[\overline{\gamma}_k - \mathbb{1}(x_0)\overline{\beta}_k - \mathbb{1}^*(x_0)\overline{\beta}_k^*\right] x_{[\mathrm{M}]} + \mathbb{1}(x_0)\overline{\beta}_k + \mathbb{1}^*(x_0)\overline{\beta}_k^*,$$

$$\text{where } \mathbb{1}(x_0) = \begin{cases} 1 & \text{if } \mathrm{argmax}\, x_0 \in [0, J), \\ 0 & \text{otherwise.} \end{cases}, \mathbb{1}^*(x_0) = \begin{cases} 1 & \text{if } \mathrm{argmax}\, x_0 \in [J, J + J^*), \\ 0 & \text{otherwise.} \end{cases},$$

$$x_{[\mathrm{M}]} = x \leftarrow \mathrm{argmax}\, x = J + J^* \text{and } \overline{\alpha}_k, \overline{\beta}_k, \overline{\beta^*}_k, \overline{\gamma}_k \text{ are the corresponding cumulative product.}$$

(11)

The reverse process predicts $\tilde{x}_0^{\mathrm{v}}$ and $\tilde{x}_0^{\mathrm{a}}$ by training denoising network $p_{\theta_1}(\tilde{x}_0^{\mathrm{v}}|x_k, y, l)$ and $p_{\theta_2}(\tilde{x}_0^{\mathrm{a}}|x_k, z_{\tilde{x}_0^{\mathrm{v}}}, l)$, respectively. Then the forward process is used to compute $p_{\theta_1}(x_{k-1}^{\mathrm{v}}|x_k, y, l)$ as expressed in Eq. (12) and $p_{\theta_2}(x_{k-1}^{\mathrm{a}}|x_k, z_{\tilde{x}_0^{\mathrm{a}}}, l)$ as expressed in Eq. (13).

$$p_{\theta_1}(x_{k-1}^{\mathrm{v}}|x_k, y, l) = \sum_{\tilde{x}_0^{\mathrm{v}}} q(x_{k-1}^{\mathrm{v}}|x_k, \tilde{x}_0^{\mathrm{v}}) p_{\theta_1}(\tilde{x}_0^{\mathrm{v}}|x_k, y, l),$$

(12)

$$p_{\theta_2}(x_{k-1}^{\mathrm{a}}|x_k, z_{\tilde{x}_0^{\mathrm{v}}}, l) = \sum_{\tilde{x}_0^{\mathrm{a}}} q(x_{k-1}^{\mathrm{a}}|x_k, \tilde{x}_0^{\mathrm{a}}) p_{\theta_2}(\tilde{x}_0^{\mathrm{a}}|x_k, z_{\tilde{x}_0^{\mathrm{v}}}, l)$$

(13)

### A.2  Perceiver Transformer

We use the Perceiver Transformer [41] to encode extremely long input sequences, which improves the computation efficiency. We maintain a set of latent vectors $z_Q$ of dimensions $\mathbb{R}^{2048 \times 256}$ which are randomly initialized. Then we compute cross attention between the input sequence and $z_Q$. The process is illustrated in Fig. 8. $z_Q$ are processed with 6 self-attention layers, and then decoded into output with the same dimension space of input via cross attention.

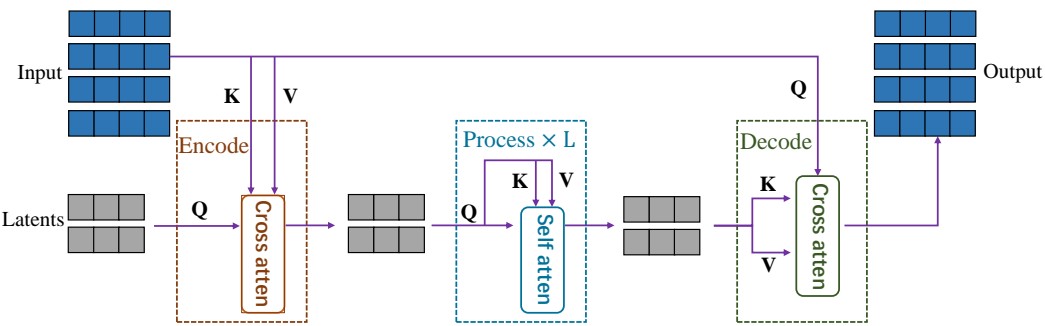

Figure 8: Illustration of Perceiver Transformer architecture. Perceiver is a latent-space transformer. Q, K, V represent queries, keys, and values, respectively. We use $L = 6$ self attention layers in our implementation.

### A.3  Implementation Details

In this section, we give the pseudocodes of pre-training and fine-tuning in Alg. 1 and Alg. 2 respectively. Then we describe the details of the training process, architecture and hyperparameters:

- Following previous works [36], we sample diffusion timestep $k \sim q(k)$ with a importance sampling strategy, where $q(k) \propto \sqrt{\mathbb{E}[\mathcal{L}_{\mathrm{vlb}}^2]}$.

- We set batch size as 20 for pre-training and 40 for fine-tuning. We train our model using Adam optimizer [49] with $2e^{-4}$ learning rate for $2e^6$ training steps.

- For pre-training, we represent our denoising network $p_{\theta_1}$ as a Perceiver Transformer described in §A.2. MLP $f_t$, which processes the language embeddings given by the CLIP encoder, MLP $f_y$, which processes the historical video tokens, and MLP $f_{\boldsymbol{x}_k}$ are 2-layered MLPs (prepended by a layer norm [3] and with Mish activation). MLP $f_{Ti}$, which processes diffusion timestep $k$, is a 2-layered MLP (prepended by a Sinusoidal embedding and with Mish activation). Finally, we use a linear layer to project the encodings given by the Perceiver Transformer into the original dimension space of the input.

- For fine-tuning, we represent our denoising network $p_{\theta_2}$ as a GPT2 Transformer. Similar to the architecture of $p_{\theta_1}$, we first use MLPs with Mish activation to project different inputs into the same hidden dimension space, and then recover the encodings given by the GPT2 Transformer into the original dimension space via a linear layer.

- We choose the sequence length $H = 4$ and $M = 1$ for future actions and videos prediction.

- We set $h = 20$ which means that we predict future videos after 20 steps.

- We set diffusion timesteps $K = 100$.

- We run all the experiments on a single RTX 3090 machine.

# B The Details of Baselines

We describe the implementation details of baselines used in our experiments as follows:

- **R3M-Diffusion**. We borrow the pre-trained R3M encoder from `https://github.com/facebookresearch/r3m`, and leverage the encoder to encode sing-view RGB images in Meta-World and multi-view RGB images in RLBench. Thus, we skip the pre-training process and directly train our discrete diffusion model on the robot data. The model architecture and hyper-parameters of R3M-Diffusion are almost the same as $p_{\theta_2}$. The encoded hidden representation encoded by R3M is denoted as $z_{R3M}$, then the denoising network can be written as $p_\theta(\boldsymbol{x}_{k-1}^{\mathrm{a}}|\boldsymbol{x}_k, z_{R3M}, \boldsymbol{l})$.

- **MTDIFF-P**. We borrow the official codes of MTDIFF-P from `https://github.com/tinnerhrhe/MTDiff`. In order to process high-dimensional images instead of low-dimensional states, we leverage the R3M encoder in R3M-Diffusion to obtain the visual representations.

- **VIDEO-MTDT**. We use the language $l$ to indicate tasks, which are encoded as a vector with size 2048 by the same CLIP encoder used in VPDD. We take the video tokens used in VPDD as the states. Then we follow the implementation from `https://github.com/kzl/decision-transformer/` to train Video-MTDT on the limited robot data.

- **VPDD-w/o-human**. During the pre-training stage of VPDD, we remove the human videos and only use the robot videos for training.

- **SODA**. Following the SODA paper [39] and open-sourced codes from `https://github.com/FutureXiang/soda`, we first maintain an encoder $E_{\mathrm{SODA}}$ equipped with the same Perceiver Transformer in $p_{\theta_1}$. Then during the pre-training stage, a representation $z$ is first encoded by $E_{\mathrm{SODA}}$ conditioning on $\boldsymbol{e}_t$ and $\boldsymbol{l}$, i.e., $z = E_{\mathrm{SODA}}(\boldsymbol{e}_t, \boldsymbol{l})$. Then $\boldsymbol{x}_{k-1}$ is obtained via the following process:

$$
\begin{aligned}
\boldsymbol{x}_{k-1} &= \mathrm{Attn}\left(\boldsymbol{x}_k, z\right) \\
\boldsymbol{x}_{k-1} &= \mathrm{Attn}\left(\boldsymbol{x}_{k-1}, \boldsymbol{x}_{k-1}\right),
\end{aligned}
\tag{14}
$$

where Attn is an attention operation [84, 55] where the queries are formed from $x$, the keys and values from $y$. The encoder is trained end-to-end and to minimize the same loss term $\mathcal{L}_{\mathrm{vlb}}$. During the fine-tuning stage, $z_{\tilde{\boldsymbol{x}}_0^{\mathrm{v}}}$ is replaced of $z$ output by $E_{\mathrm{SODA}}$. The model architecture and other hyper-parameters during fine-tuning remain the same.

---

**Algorithm 1** Pre-Training Stage of VPDD

---

**Initialize**: given unified transition matrix $\{\boldsymbol{Q}_k\}$, well-trained VQVAE, training iterations $N$, initial network parameters $\theta_1$ and $\theta_2$, learning rate $\eta$.

1: **for** video clips $v_t$ in $\{\tau_i\}$ **do**
2:    $\boldsymbol{e}_t \leftarrow$ VQVAE-Encoder$(v_t)$
3: **end for**
4: **for** $n = 1$ **to** $N$ **do**
5:    Sample a batch $\mathcal{B} = (\boldsymbol{x}^v, \boldsymbol{x}^a, \boldsymbol{y}, \boldsymbol{l})$ from human and robot videos, where $\boldsymbol{x}^a \leftarrow \phi$
6:    Sample diffusion timestep $k$ from $[1, K]$ with importance sampling strategy
7:    $\boldsymbol{x}_k \leftarrow q(\boldsymbol{x}_k|\boldsymbol{x}_0)$, where $\boldsymbol{x}_0 = [\boldsymbol{x}^v, \boldsymbol{x}^a]$   $\triangleright$ Eq. (11) and (4)
8:    $\mathcal{L}_{\text{vlb}} = \begin{cases} \mathcal{L}_0 & \text{if } k = 1, \\ \mathcal{L}_{k-1} & \text{otherwise.} \end{cases}$  $\triangleright$ Eq. (6)
9:    $\theta_1 \leftarrow \theta_1 - \eta\nabla_{\theta_1}\mathcal{L}$   $\triangleright$ Adam optimizer
10: **end for**

---

---

**Algorithm 2** Fine-Tuning Stage of VPDD and Evaluation

---

*# Fine-Tuning Process*
**Initialize**: given unified transition matrix $\{\boldsymbol{Q}_k\}$, well-trained VQVAE, training iterations $N$, pre-trained network parameters $\theta_1$ and initial parameters $\theta_2$, learning rate $\eta$.

1: **for** video clips $v_t$ and actions $a_t$ in $\{\tau_i\}$ **do**
2:    $\boldsymbol{e}_t \leftarrow$ VQVAE-Encoder$(v_t), a_t \leftarrow$ Discretize$(a_t)$
3: **end for**
4: **for** $n = 1$ **to** $N$ **do**
5:    Sample a batch $\mathcal{B} = (\boldsymbol{x}^v, \boldsymbol{x}^a, \boldsymbol{y}, \boldsymbol{l})$ from videos and discretized actions dataset.
6:    Sample diffusion timestep $k$ from $[1, K]$ with a importance sampling strategy
7:    $\boldsymbol{x}_k \leftarrow q(\boldsymbol{x}_k|\boldsymbol{x}_0)$, where $\boldsymbol{x}_0 = [\boldsymbol{x}^v, \boldsymbol{x}^a]$   $\triangleright$ Eq. (11) and (4)
8:    $\mathcal{L}_{\text{vlb}} = \begin{cases} \mathcal{L}_0 & \text{if } k = 1, \\ \mathcal{L}_{k-1} & \text{otherwise.} \end{cases}$  $\triangleright$ Eq. (6)
9:    $\theta_2 \leftarrow \theta_2 - \eta\nabla_{\theta_2}\mathcal{L}$   $\triangleright$ Adam optimizer
10: **end for**

*# Evaluation Process*
1: Given a task, reset the environment
2: Obtain the initial video clips $v_0$, language instructions $\boldsymbol{l}$
3: **for** $t = 0$ **to** $t_{\max}$ **do**
4:    Initialize $[\boldsymbol{x}_K^v, \boldsymbol{x}_K^a] \sim p(\boldsymbol{x}_K)$   $\triangleright$ Eq. (10)
5:    Construct $\boldsymbol{y} \leftarrow \boldsymbol{e}_t =$ VQVAE-Encoder$(v_t)$
6:    **for** $k = K$ **to** $1$ **do**
7:      Sample $\boldsymbol{x}_{k-1}^v \sim p_{\theta_1}(\boldsymbol{x}_{k-1}^v|\boldsymbol{x}_k, \boldsymbol{y}, \boldsymbol{l})$ and obtain $z_{\tilde{\boldsymbol{x}}_0^v}$ from $p_{\theta_1}$ encoding $\triangleright$ Eq. (12)
8:      Sample $\boldsymbol{x}_{k-1}^a \sim p_{\theta_2}(\boldsymbol{x}_{k-1}^a|\boldsymbol{x}_k, z_{\tilde{\boldsymbol{x}}_0^v}, \boldsymbol{l})$   $\triangleright$ Eq. (13)
9:      $\boldsymbol{x}_{k-1} = [\boldsymbol{x}_{k-1}^v, \boldsymbol{x}_{k-1}^a]$
10:    **end for**
11:    Obtain predicted videos via VQVAE-Decoder$(\boldsymbol{x}_0^v)$
12:    Reconstruct executable action sequence $[a_t, \cdots, a_{t+H-1}]$ from $\boldsymbol{x}_0^a$
13:    Execute the first action $a_t$ as the current action to interact with the environment
14:    Obtain the next observed image(s), and update $v_t$
15: **end for**

---

## C   Datasets and Experiments

### C.1   Dataset

**Meta-World.** We use the official codes from `https://github.com/Farama-Foundation/Metaworld` to collect 20 expert demonstrations for each task in Meta-World. The image size

is $260 \times 260$. The dimension of action is 4, representing the 3D position movements of the end effector and the variation of gripper openness. Every demonstration collected has 150 time steps.

**RLBench.** We use the official codes from `https://github.com/peract/peract` to generate a dataset for RLBench via a motion planner. Each demonstration consists of multi-view RGB images (i.e., front, left shoulder, right shoulder and wrist) and 8-dimensional actions including 3-dimensional transitions, 4-dimensional quaternion and a binary value about gripper openness. Following prior works [42, 75], we extract macro actions (keyframe actions) from collected demonstrations and leverage networks to predict the keyframe actions instead of consistent continuous actions. Specifically, a set of keyframe actions $\{k_1, k_2, \cdots, k_m\} \subset \mathcal{A}$ is captured with a simple heuristic: an action is a keyframe if (1) the joint velocities are near zero and (2) the gripper open state has not changed. Each data point in the demonstration $\tau$ can then be cast as a "predict the next (best) keyframe action" task [44]. In this way, the sequence length of actions that need to be predicted is significantly reduced from hundreds of small steps to typically less than 10 macro steps.

## C.2 Task Details

| Task | Variation Type | # of Variations | Language Template |
|---|---|---|---|
| slide block | color | 4 | "slide the block to __ target" |
| sweep to dustpan | size | 2 | "sweep dirt to the __ dustpan" |
| meat off grill | category | 2 | "take the __ off the grill" |
| turn tap | placement | 2 | "turn __ tap" |
| put in drawer | placement | 3 | "put the item in the __ drawer" |
| close jar | color | 20 | "close the __ jar" |
| drag stick | color | 20 | "use the stick to drag the cube onto the __ target" |
| stack blocks | color, count | 60 | "stack __ __ blocks" |
| screw bulb | color | 20 | "screw in the __ light bulb" |
| put in safe | placement | 3 | "put the money away in the safe on the __ shelf" |
| place wine | placement | 3 | "stack the wine bottle to the __ of the rack" |
| put in cupboard | category | 9 | "put the __ in the cupboard" |
| push buttons | color | 50 | "push the __ button, [then the __ button]" |
| insert peg | color | 20 | "put the ring on the __ spoke" |
| stack cups | color | 20 | "stack the other cups on top of the __ cup" |
| place cups | count | 3 | "place __ cups on the cup holder" |

Table 3: Language-Conditioned Tasks in RLBench [43] with various variations.

We take Meta-World as a main benchmark to evaluate our method and baselines, which consists of 50 diverse manipulation tasks. The poses and positions of goals are randomly generated during evaluation. These tasks require an agent to identify the observed sing-view RGB images and reach the goals with the correct behavior. See Fig. 9 for a sample visualization of the tasks.

We select 16 tasks out of 100 tasks from RLBench [43] that involve at least two or more variations to evaluate the multi-task and generalization capabilities of agents. Task variations include randomly sampled colors, sizes, counts, placements, and categories of objects. The set of colors include 20 instances: `colors` = {`red`, `maroon`, `lime`, `green`, `blue`, `navy`, `yellow`, `cyan`, `magenta`, `silver`, `gray`, `orange`, `olive`, `purple`, `teal`, `azure`, `violet`, `rose`, `black`, `white`}. The set of sizes include 2 instances: `sizes` = {`short`, `tall`}. The set of counts include 3 instances: `counts` = {`1`, `2`, `3`}. The placements and object categories are specific to each task. For instance, `put in cupboard` includes 9 YCB objects. In addition to these semantic variations, objects are placed on the tabletop at random poses. The tasks in RLBench require an agent to process multi-view RGB images properly and generalize to different variations that could be unseen in training. The details of variations for each task are referred to Table 3.

## C.3 Experimental Setup and Additional Results

**Generalize to unseen scenes.** To evaluate the generalization ability of our method, we change the camera position and table texture in the Meta-World benchmark to generate out-of-distribution observations. Borrowing the official codes from `https://github.com/RLAgent/factor-world`,

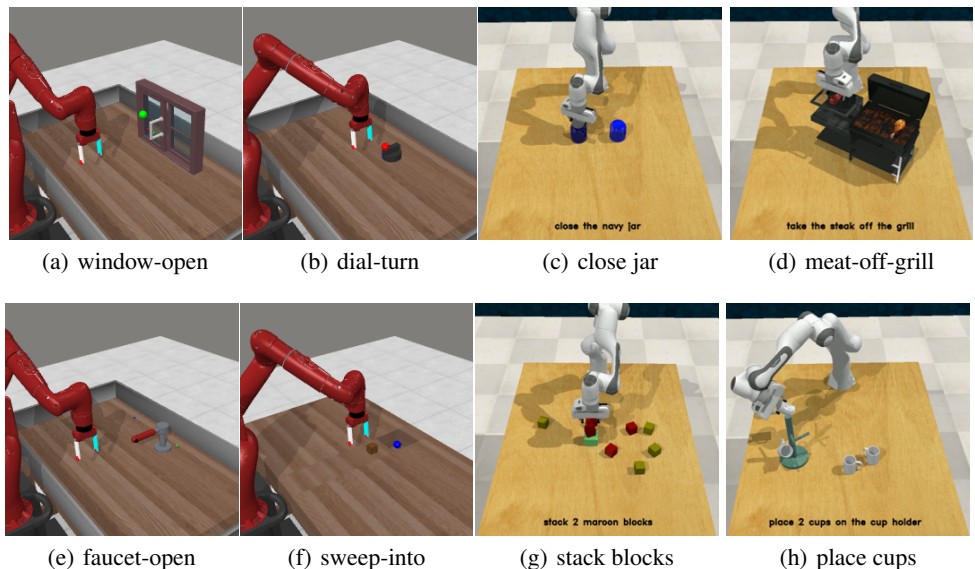

| (a) window-open | (b) dial-turn | (c) close jar | (d) meat-off-grill |
| (e) faucet-open | (f) sweep-into | (g) stack blocks | (h) place cups |

Figure 9: Visualization of several tasks in Meta-World and RLBench.

| Unseen Tasks | VPDD | VPDD (oracle) |
|---|---|---|
| hand-insert-v2 | 32% | 36% |
| bin-picking-v2 | 78% | 84% |
| door-unlock-v2 | 100% | 100% |

Table 4: Average success rate of VPDD on unseen tasks.

we set the camera at another corner position and generate unseen table texture randomly while rendering. See Fig. 10 for visualization.

**Generalize to unseen tasks.** In the following, we show that VPDD can generalize on different tasks from the same domain. In experiments, during stage 2, we train the VPDD on 47 tasks on MetaWorld and leave 3 unseen tasks to test the generalizability. In stage 3, we fine-tune the pretrained model on these 3 tasks. We report the success rate of the final model trained over 0.2M gradient steps. The following experimental results in Table C.3 demonstrate the promising generalization ability of our model, as the performance gap compared with the oracle is very small.

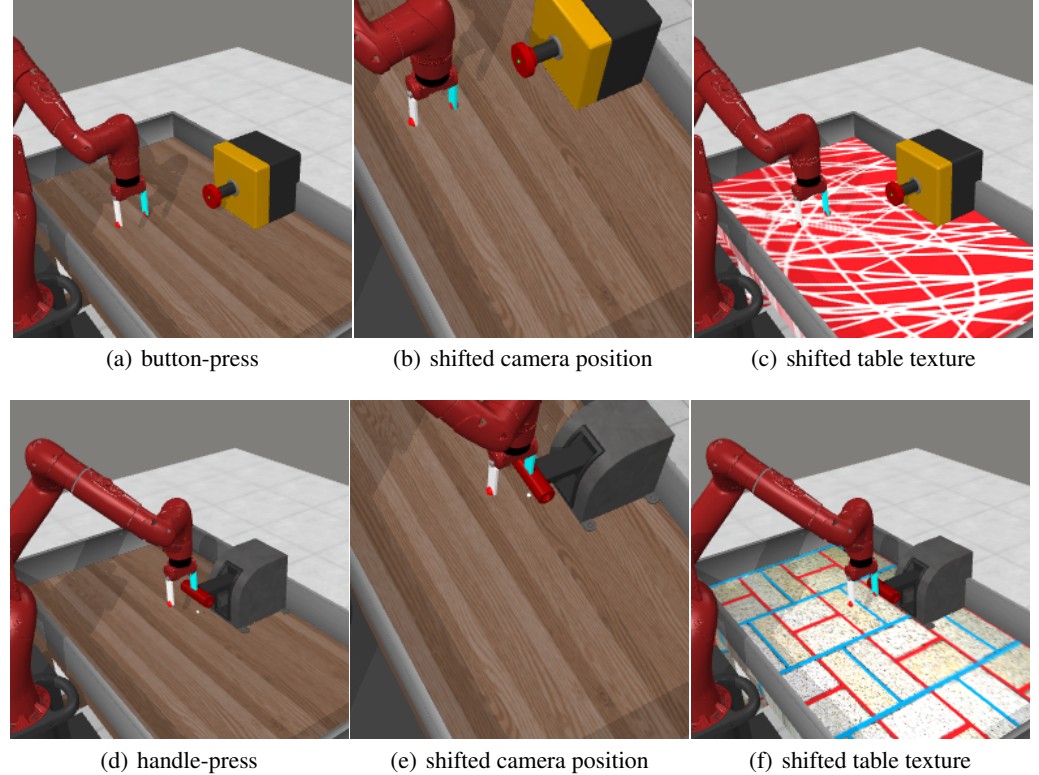

(a) button-press      (b) shifted camera position      (c) shifted table texture

(d) handle-press      (e) shifted camera position      (f) shifted table texture

Figure 10: A visualized example of shifted camera position and table texture on *button-press-v2* and *handle-press-v2* tasks. Table texture is generated randomly during evaluation.

## D   Limitations and Future Work

We present illustrative samples of robot videos generated using discrete diffusion model $p_{\theta_1}$ in Fig. 3 and Fig. 11. More samples are available at `https://video-diff.github.io`. It is noteworthy that VPDD is able to generate dynamic consistent future videos, incorporating information beneficial for low-level control while maintaining coherence across distinct perspectives. However, it is imperative to acknowledge that our primary contribution is not on generating high-quality videos with exceptional resolution and meticulous semantic details. Consequently, some blurriness may be observed in our predicted videos, exemplified by deviations in the gripper's pose. See Fig. 12 for a failure example.

For future work, we could consider enhancing the coherence of videos across diverse views by leveraging the recently released Ego-exo4d data [28]. This extension encompasses considerations beyond solely temporal dynamics. Moreover, for the augmentation of video quality and the optimization of decision-making performance, it is worth exploring to scale up both the training dataset and model capacity.

## E   Broader Impact

This paper presents work whose goal is to advance the field of Machine Learning. Specifically, we propose a novel pretraining-finetuning framework to make better use of a copious amount of human actionless videos. Since this method is easy to reproduce (as we will release our code soon) and exhibits the SOTA performance, it encourages future research to further advance this field. There are many potential societal consequences of our work, none of which we feel must be specifically highlighted here.

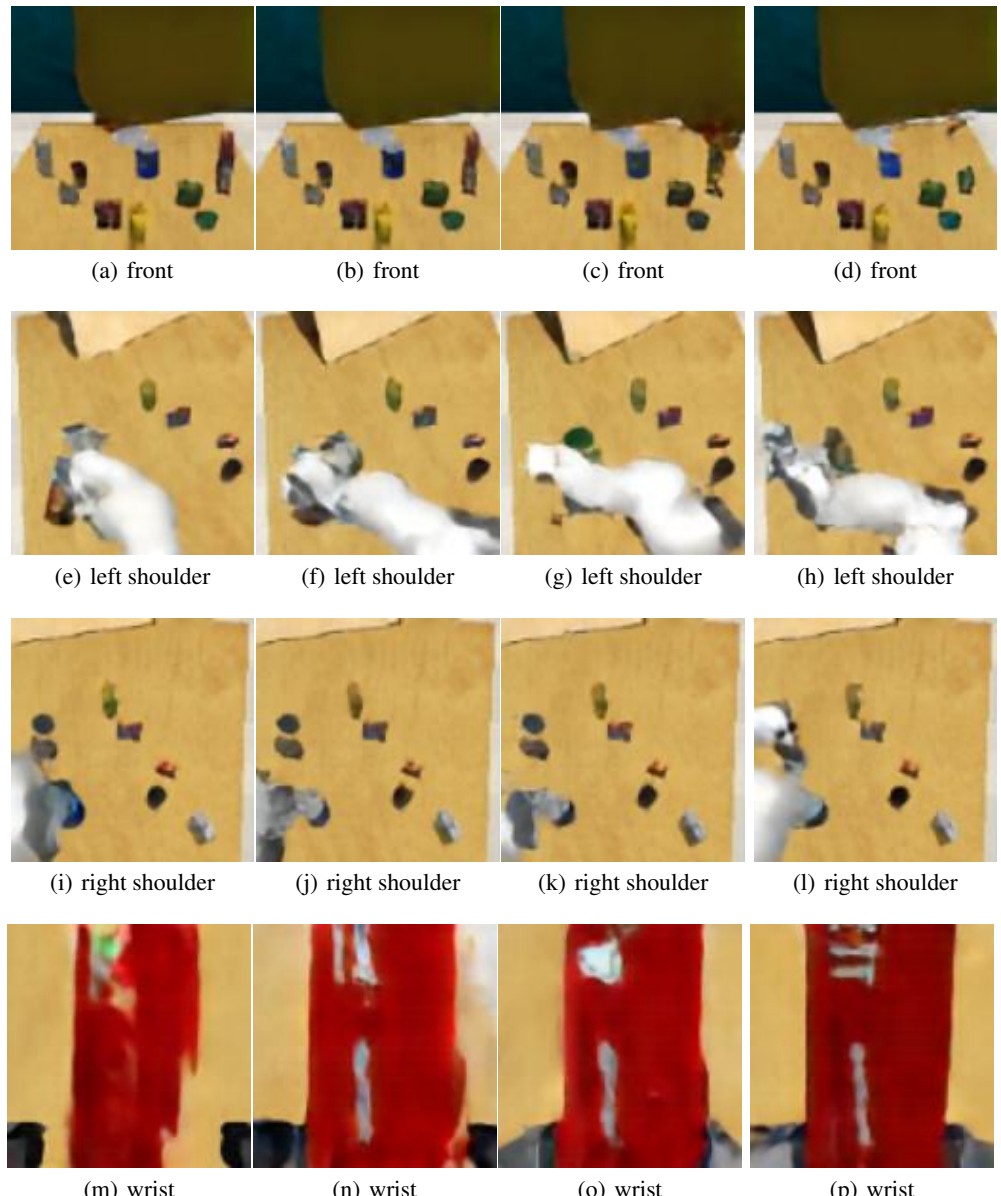

(a) front      (b) front      (c) front      (d) front

(e) left shoulder    (f) left shoulder    (g) left shoulder    (h) left shoulder

(i) right shoulder    (j) right shoulder    (k) right shoulder    (l) right shoulder

(m) wrist      (n) wrist      (o) wrist      (p) wrist

Figure 11: Predicted video given by $p_{\theta_1}$ for task *put the crackers in the cupboard*.

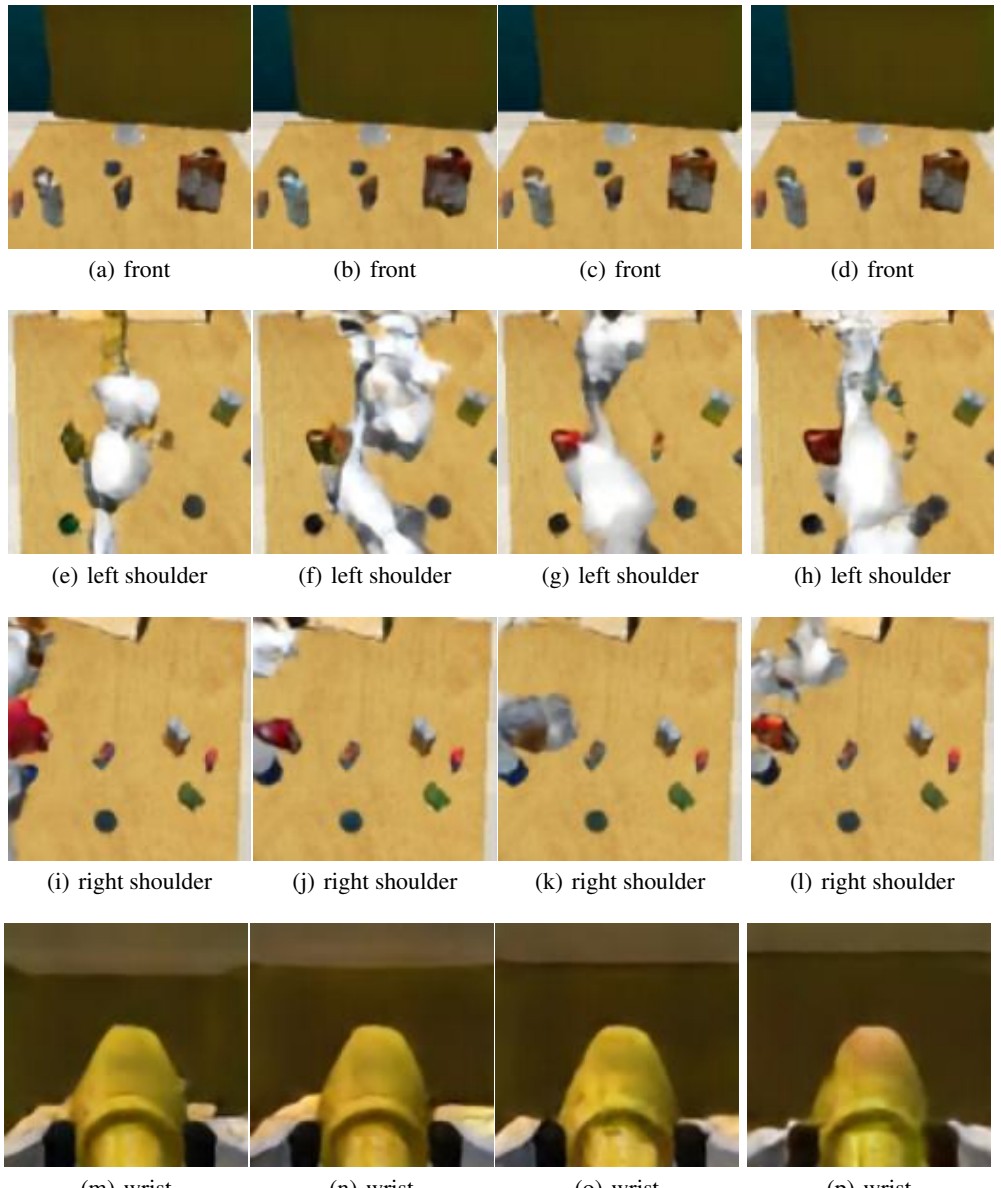

Figure 12: Predicted video given by $p_{\theta_1}$ for task *put the mustard in the cupboard*. The pose of the robotic arm in the video is somewhat blurry, with deficiencies in correspondence across different views.

