# OpenReview forum: "Learning an Actionable Discrete Diffusion Policy via Large-Scale Actionless Video Pre-Training"
_NeurIPS.cc/2024/Conference — NeurIPS 2024 poster_

### Official Review · Reviewer_wSvy · 2024-07-12

**Soundness:** 3
**Presentation:** 4
**Contribution:** 3
**Rating:** 6
**Confidence:** 5

**Summary:**

The authors propose an approach that takes advantage of human-object interaction videos to improve policies. Their proposal involves three steps: 1) learn a video tokenizer common to both human and robot videos. 2) Learn a discrete-diffusion model that performs video token denoising on both human and robot videos. 3) Learn a discrete-diffusion model that performs action denoising on only robot data. They show this technique results in better performance as compared to baselines.

**Strengths:**

1. Proposed idea is very interesting. It brings together ideas from self-supervised pre-training and diffusion policies.
2. They achieve strong performance on a number of manipulation benchmarks outperforming many baselines.
3. Clear explanation of the methods.

**Weaknesses:**

1. The paper claims that pre-training on a large dataset of human-object interaction videos without actions is useful for learning actions for robots. However from the experiments it is not clear how much the human videos are adding to the policy performance? There is a significant amount of data difference between the models trained with or without human videos. Yet the performance between the two models is close. How does the model without human data perform given some more robot trajectories? In Fig. 1, for comparison without human videos the drop in performance in marginal. In Fig. 5 seems there is a bigger drop in performance which might be compensated by having multiple camera views in the robot training data.

2. Ego4D videos are very different visually from the tasks and environments shown. It is not clear which tasks from Ego4D are helping in transferring good features for the robotics tasks. How much of the video tokenizer codebook is shared between the human and robot videos. Because the dataset statistics look so different I wonder if enough of the codebook is being shared by the two datasets?

3. Generating egocentric videos is challenging. If the model is good at generating egocentric videos then that in itself is a big contribution. What do the generated videos look like for the Ego4D dataset? Does performance on this task correlate with better downstream policy performance or is the fine-tuning step capable of recovering good policies on video prediction models that are not that good?

4. Another baseline to compare against would be a VideoMAE[1] kind of method to learn the video features on just robot dataset and then do diffusion on that. This will be a good way to make the case for a tokenizer based diffusion model to learn video features.

[1] VideoMAE: Masked Autoencoders are Data-Efficient Learners for Self-Supervised Video Pre-Training. Zhan Tong, Yibing Song, Jue Wang, Limin Wang

**Questions:**

See above weakness section.

**Limitations:**

Yes.

---

> ### Author Rebuttal · Authors · 2024-08-07
>
> Thank you for your thorough comments and valuable suggestions! Below, we carefully address your concerns:
> >1. ... how much the human videos are adding to the policy performance? ... How does the model without human data perform given some more robot trajectories?...
>
> Thanks for your question.
>
> (1) Regarding the effectiveness of human video, we highlight that since the robot videos are used in pretraining, the predicted videos can directly benefit policy fine-tuning. As a result, even without using human data, VPDD remains a strong baseline as the video predictor is reliable for predicting such "in-distribution" scenes.
>
> (2) For experiments in unseen scenes, since the pretraining dataset does not contain videos with the same backgrounds or camera views, VPDD relies heavily on the generalization ability learned from large-scale human datasets. The superior performance in such unseen scenes verifies that large-scale human-data pretraining extracts commonsense knowledge and representations that can be shared between human and unseen robot scenarios. **Furthermore, we ablate the number of human videos used during pre-training and show the performance changes in Figure 1 of the global pdf**. We find that more human videos help increase the success rates, especially boosting the generalizability of unseen scenes.
>
> (3) It is true that adding additional robot data in these unseen views will improve the performance without human-data pretraining. However, we cannot expect to collect all kinds of variations in robotic tasks, such as visual appearances, camera views, lighting changes, scene structures, embodiments, etc. This is a strong assumption and generally unrealistic. Large-scale human interaction data provides a desirable choice, as it learns a prior of visual representation and dynamic rules to facilitate effective generalization to unseen scenes.
>
>
>
> >2. ...How much of the video tokenizer codebook is shared between the human and robot videos...
>
> Thanks for your interesting question. We would like to clarify that although Ego4d videos and robot videos show different visulization, they share substantial knowledge which can benefit robot learning. For example, (1) Human follows the dynamic rules of the physical world to interact with the objects, which is similar to how a robot system interacts with the environment. (2) The human videos cover a wide range of objects, visual features, and tasks, aiding robots in understanding and adapting to real-world situations and facilitating knowledge transfer and versatility in robot systems. As a result, it is not necessary for human videos to come from similar tasks and views. For example, although human videos are egocentric, they can still benefit the prediction of multi-view videos (left, right, front, and overhead), like those in RLBench.
>
> Following your advice, we sample 50 clips from both human and robot videos and then computed the shared VQ-VAE codes among them. We find that 1446 codes are shared, accounting for 70.6% of the codebook (with a size of 2048).
>
> To further validate the knowledge transfer from human to robot, we include an ablation study that excludes robot data during pre-training. **The results presented in Table 1 of the global PDF indicate that VPDD-w/o-robot still maintains a satisfactory level of success rates and remains competitive compared to the baselines**. Specifically, for the 'drawer-open' task, which shares common behaviors and skills seen in human videos, this variant achieves a 100% success rate.
>
> >3. ...What do the generated videos look like for the Ego4D dataset? Does performance on this task correlate with better downstream policy performance or is the fine-tuning step capable of recovering good policies on video prediction models that are not that good?
>
> (1) We present the video prediction results of the Ego4d dataset on our project website. VPDD can also generate dynamically consistent future videos for the human domain.
>
> (2) It is expected that better video prediction will result in improved downstream policy performance, as the representation $z_{\tilde{{x}}_{0}^{\rm v}}$ provided to the fine-tuning stage will be more informative. **As shown in Figure 1 of the global PDF, we observe that VPDD's performance improves as more human videos are added**. One important reason is that video prediction performance improves with more data seen during pre-training.
>
> (3) Moreover, our method can also recover good policies on video prediction models that are not that good. We exclude robot videos in pre-training, which could lead to worse performance in robot video prediction. **As shown in Table 1 of the global pdf, VPDD can still achieve a satisfactory success rate**.
>
>
> >4. ... Another baseline to compare against would be a VideoMAE[1] kind of method to learn the video features on just robot dataset ...
>
> Thanks for your suggestion. We finetune VideoMAE (ViT-Base pretrained on Kinetics-400) using our robot dataset and its official open-sourced codes, and then leverage the learned video features to learn actions like R3M-Diffusion baseline. We evaluate its final performance on 20 episodes per task in Metaworld. The average success rate is 46%, which is similar to the VC-1 diffusion baseline, which is also a ViT trained by MAE. However, VPDD still outperforms this baseline by about 23% performance improvement.
>
> Overall, we sincerely thank you for reviewing our paper and for your constructive suggestions again. We hope we have resolved all the concerns. We would appreciate it if you could kindly consider raising the score. We are always willing to address any of your further concerns. Thank you!

---

> > ### Comment · Reviewer_wSvy · 2024-08-10
> >
> > Thank you for answering the questions and conducting the new experiments. I will be raising my score to weak accept as my concerns are addressed. Please add these results in the paper to provide the readers more context.

---

> > > ### Author Response · Authors · 2024-08-11
> > > **Thank you for raising the score!**
> > >
> > > Thank you for raising your score and for supporting our work! We will incorporate the reviewers' valuable suggestions and the added experiments into the next version of our paper.

---

### Official Review · Reviewer_ew7s · 2024-07-12

**Soundness:** 4
**Presentation:** 3
**Contribution:** 3
**Rating:** 6
**Confidence:** 4

**Summary:**

In this paper, the authors focus on the problem of policy learning via leveraging large video data without action labels. To this end, they employ a discrete diffusion framework which is first employed to predict future quantized video frames. Afterwards, the model is fine-tuned on robot data to learn the final policy. The authors evaluate their approach on the Meta-World and RLBench and demonstrate promising quantitative results over other approaches.

**Strengths:**

1. The paper addresses a very timely problem: Most available video data has no action labels. A major question is how to properly leverage this data to learn general policies for control tasks.

2. For an approach which utilizes a pre-training and a fine-tuning stage with frozen networks, the approach isn't overly complex. It's simple relative to other approaches. It therefore should be easy for researchers to implement and reproduce results.

3. Quantitative results for both Meta-World and RLBench are strong over baselines. The results in are quite promising.

**Weaknesses:**

1. Technical novelty is limited. The approach is similar to a lot of other approaches: quantize videos using VQVAE, predict future tokens using transformer, somehow account for missing action tokens.

2. The scope of domains for evaluation domains is somewhat limited. It would be interesting to see if the results hold on other robotic datasets.

**Questions:**

1. While I believe the current results in the paper are convincing, it still may strengthen the paper to demonstrate results on even more datasets.

**Limitations:**

Yes, the authors have properly addressed limitations.

---

> ### Author Rebuttal · Authors · 2024-08-07
>
> Thank you for your thorough review and for a positive assessment of our work. Here, we carefully address your concerns as follows:
>
> >1. Technical novelty is limited. The approach is similar to a lot of other approaches: quantize videos using VQVAE, predict future tokens using transformer, somehow account for missing action tokens.
>
> Thanks for the comment. While some modules in our method are adapted from previous works, we have innovatively applied and carefully designed these architectures within the novel context of video-based robot decision-making, thus yielding significant contributions to this field. Additionally, We would like to highlight that the main contribution of our paper lies in the proposed pretraining-finetuning paradigm for efficient policy learning with a limited number of robot demonstrations. We achieve this through pre-training on large-scale actionless human videos and finetuning the action-prediction parameters on robot data. Through this paradigm, we can develop a generalist agent capable of processing multimodal information as inputs and performing multiple tasks using limited robot data, which holds significant importance for the community.
>
> We **are the first** to formulate both video prediction and action learning processes as unified discrete denoising problems, enabling the learning of different modalities (videos in pre-training and actions in fine-tuning) across different stages. Notably, our paper also provides insights to the community, including methods for bridging the gap between human and robot domains by representing videos as discrete latent codes and enhancing policy learning with foresight from predicted videos.
>
> >2. The scope of domains for evaluation domains is somewhat limited. It would be interesting to see if the results hold on other robotic datasets.
>
> Thanks for your suggestion! To further validate the superior performance of our method, we train our model using 400 episodes from Language Table [1] blocktoblock dataset (400 out of 200,000 episodes), and finetune the pretrained Octo-Base model using the same dataset for $1e5$ steps. The Language Table is part of the Open-X dataset and is an extremely challenging benchmark due to the random instructions and block locations in each episode. We evaluate their performance on the Language Table environment over 100 episodes. We find that Octo-Base gets a **0%** success rate while our model can achieve a **10%** success rate. Our findings are consistent with a recent work [2].
>
> [1] Corey Lynch et al., Interactive Language: Talking to Robots in Real Time, Arxiv 2022.
>
> [2] W. Zhao et al., VLMPC: Vision-Language Model Predictive Control for Robotic Manipulation, RSS 2024.
>
> We hope to have addressed all the raised concerns and would be happy to respond to further questions and suggestions. Thank you again for your time and efforts!

---

> > ### Author Response · Authors · 2024-08-12
> > **Respectful Reminder**
> >
> > Dear Reviewer,
> >
> > We hope this message finds you well. Since the discussion phase will end in approximately two days, we want to kindly follow up on the response we provided to your valuable comments and questions. Your feedback has been crucial in refining our work, and we greatly appreciate the time and effort you have invested in reviewing our paper. If you have any additional thoughts or questions about our responses, we would be more than happy to address them.
> >
> > Thank you once again for your insightful review! We look forward to your further feedback.

---

> > ### Comment · Reviewer_ew7s · 2024-08-12
> > **Response to Rebuttal**
> >
> > I would like to thank you for addressing my concerns. I stand by my rating. I believe this paper is an interesting contribution to the conference and would like to see it accepted.

---

> > > ### Author Response · Authors · 2024-08-13
> > >
> > > Thank you again for your time and efforts in reviewing our paper! We are glad to see you recognize our contribution. We will incorporate the reviewers' valuable suggestions and the added experiments into the next version of our paper.

---

### Official Review · Reviewer_kd9D · 2024-07-13

**Soundness:** 3
**Presentation:** 3
**Contribution:** 2
**Rating:** 6
**Confidence:** 4

**Summary:**

The authors propose utilizing large scale internet videos along with robot data to train a diffusion model which predicts video future frames conditioned on past video frames and language description of videos. This diffusion model operated on the vector quantized embeddings of the video frames. Subsequently, this diffusion model is adapted for action generation which is trained using a smaller amount of robot video data with action labels. The analysis is performed on the Metaworld suite and RLBench.

**Strengths:**

1. The paper is well written, clear and easy to understand
2. The idea is well motivated. The question of utilizing internet scale knowledge for embodied intelligence is an open area of research.
3. The experiment results are statistically significant with multiple runs reported accompanied by error bars.
4. The method performs well in comparison to baselines.

**Weaknesses:**

1. **Missing Baselines:** How does the method compare against other generalist imitation learning methods like Octo, RT-1 etc? It's unclear how such methods compare in the finetuning setting.
2. **Missing Ablations on amount of robot data used during pretraining:** While no action labels have been used during pretraining, the model still has access to robot trajectories. How does the method perform when no robot data is present during pretraining?
3. **Lack of robot experiments:** Does the model extend to any real robot setups?

**Questions:**

1. What are the computational costs of training the backbone?
2. Are the results on RLBench and Metaworld finetuned for every task or are in multi-task setting, i.e., do you have one model that yields the aggregate result on Metaworld or do you have one model per task?

**Limitations:**

1. The method is currently limited to simulation and it remains to be seen if the results will transfer to real robots.
2. The method still requires 20 demonstrations per task during pretraining. It’s not clear if this is a kosher assumption. What if behavior cloning was used on these 20 demonstrations? Would this yield higher results? If so, its unclear why one would use VPDD?

---

> ### Author Rebuttal · Authors · 2024-08-07
>
> Thank you for your detailed feedback and for a positive assessment of our work! We carefully address your concerns as follows.
> >1. How does the method compare against other generalist imitation learning methods like Octo, RT-1 etc?
>
> Thanks for your suggestions. We would like to clarify that Octo and RT-1 are directly trained on large-scale robot data with action labels (e.g., Octo is trained on 800K episodes from the Open X-Embodiment dataset). In contrast, our method contains a human pretraining stage that leverages actionless video pretraining, significantly reducing the requirement of action-labeled robot data (i.e., only 1000 episodes for 50 tasks in Meta-World and 160 episodes for 16 tasks in RLBench). As a result, we believe the contributions of VPDD and Octo/RT-1 are orthogonal, where VPDD focuses on leveraging actionless pretraining and few-shot finetuning, while Octo/RT-1 focuses on directly learning generalized actionable policy from large-scale robot data. Additionally, we note that Octo is a nearly concurrent work, and neither Octo nor RT-1 presents evaluation results on Meta-World and RLBench.
>
> To further validate the superior performance of our method, we train our model using 400 episodes from Language Table [1] blocktoblock dataset (400 out of 200,000 episodes), and finetune the pretrained Octo-Base model using the same dataset for $1e5$ steps. The Language Table is part of the Open-X dataset and is an extremely challenging benchmark due to the random instructions and block locations in each episode. We evaluate their performance in the Language Table environment over 100 episodes.  We find that Octo-Base gets a **0%** success rate while our model can achieve a **10%** success rate. Our findings are consistent with a recent work [2].
>
> [1] Corey Lynch et al., Interactive Language: Talking to Robots in Real Time, Arxiv 2022.
>
> [2] W. Zhao et al., VLMPC: Vision-Language Model Predictive Control for Robotic Manipulation, RSS 2024.
>
>
> >2. Missing Ablations on amount of robot data used during pretraining: While no action labels have been used during pretraining, the model still has access to robot trajectories. How does the method perform when no robot data is present during pretraining?
>
> Thanks for your suggestion. We would like to clarify that the main motivation of our paper is to make the best use of the actionless video data for downstream decision making, rather than assuming the robot videos are inaccessible. However, it is reasonable to expect that using less robot data in the pre-training stage will result in worse model performance. This is because the representation $z_{\tilde{{x}}_{0}^{\rm v}}$ provided for finetuning would be less accurate and contain noise that affects the performance. To evaluate the performance of VPDD without seeing robot data during pre-training, we add an experiment where no robot data is used in the pre-training stage, and test its performance on five tasks in Metaworld. **See the results in Table 1 of the global pdf**. We observe that while the performance of VPDD-w/o.-robot decreases, it remains competitive compared to the baselines.
>
> >3. Lack of robot experiments: Does the model extend to any real robot setups?
>
> Due to the resource limitation, we do not evaluate VPDD in real-world manipulation tasks. Nevertheless, the chosen simulation benchmarks in our paper are realistic and challenging, especially the 3D manipulation tasks with a Franka arm from RLBench. These tasks are also widely used in previous robot learning works and make it easy to compare VPDD to other baseline methods. To verify that our method can handle real-world scenarios, we fine-tune VPDD on real robot data obtained from RoboAgent data [1]. We observe that our model can quickly adapt to this domain and generate high-fidelity future videos in this context. We provide the predicted videos on the website video-diff.github.io. Additionally, we'd like to clarify that while real-world experiments can indeed be valuable, they are not a mandatory requirement for acceptance at NeurIPS.
>
> [1] Bharadhwaj H, Vakil J, Sharma M, et al. Roboagent: Generalization and efficiency in robot manipulation via semantic augmentations and action chunking. Conference on Robot Learning (CoRL). 2023.
>
> >4. What are the computational costs of training the backbone?
>
> Thank you for the question. In our experiments, we use 8 RTX 3090 GPUs to train the VQ-VAE for video tokenization over a period of 3 days. The pre-training process takes approximately 2.5 days per RTX 3090, while fine-tuning requires about 1.5 days per RTX 3090. The training time could be reduced by utilizing more advanced GPUs and employing parallel computation.
>
> >5. Are the results on RLBench and Metaworld finetuned for every task or are in multi-task setting, i.e., do you have one model that yields the aggregate result on Metaworld or do you have one model per task?
>
> Thanks for the question. They are finetuned in a multi-task setting, which leverage a single model to generate aggregated results in all downstream tasks.
>
> >6. The method is currently limited to simulation and it remains to be seen if the results will transfer to real robots.
>
> Please see the answer to Q3.
>
> >7. The method still requires 20 demonstrations per task during pretraining. It’s not clear if this is a kosher assumption. What if behavior cloning was used on these 20 demonstrations? Would this yield higher results? If so, its unclear why one would use VPDD?
>
> Thanks for the question. We have included the behavior cloning results without video prediction. As shown in Figure 4, MTDiff-p is a diffusion model that performs behavior cloning on 20 demonstrations pre-task. Our method outperforms MTDiff-p by 17.98\%.
>
> We hope to have addressed all the raised concerns and would be happy to respond to further questions and suggestions. Thank you again for your time and efforts!

---

> ### Author Response · Authors · 2024-08-12
> **Respectful Reminder**
>
> Dear Reviewer,
>
> We hope this message finds you well. Since the discussion phase will end in approximately two days, we want to kindly follow up on the response we provided to your valuable comments and questions. Your feedback has been crucial in refining our work, and we greatly appreciate the time and effort you have invested in reviewing our paper. If you have any additional thoughts or questions about our responses, we would be more than happy to address them.
>
> Thank you once again for your insightful review! We look forward to your further feedback.

---

### Official Review · Reviewer_v4Bj · 2024-07-16

**Soundness:** 3
**Presentation:** 3
**Contribution:** 3
**Rating:** 7
**Confidence:** 4

**Summary:**

The paper targets at transferring the knowledge from human videos to robotic manipulation policy. The paper proposes VPDD, a method that first pre-trains on video generation on both human and robot videos and then fine-tunes on robotic manipulation data with action labels. The intuition is, by learning to generate human and robot videos, the model obtains a "prior" (e.g., task planning, trajectory planning, etc.) of robotic manipulation which makes it easier to learn diverse manipulation tasks with even few-shot data. The authors compare VPDD to several robotic pre-training baselines and show non-trivial improvements over several benchmarks.

**Strengths:**

Clear motivation that robot manipulation data is scarce and utilizing the large amount of human data will be helpful. The design of the model and the training pipeline is sensible and the results look promising.

**Weaknesses:**

1. The pre-training stage is trained jointly on human and robot data. The amount of human data is pretty abundant here but how many robot videos are we using here? If the robot videos are too few, will it affect the resulting pre-trained model from this stage? If the pre-training stage still needs a lot of robot videos, then it still faces the problem of lacking enough robot data. On the other hand, if the pre-trained model can still generalize pretty well without robot data in the pre-trianing stage, i.e., pre-training on human data only and fine-tuning on a small amount of robot data can work pretty well, it would be very helpful.

2. Since the main message of the paper is training on human data can help learning robotic manipulation policies, it would be good to see how the performance changes when trained on different amounts of human data, for example, a scaling curve with x-axis being the number of human videos and y-axis being the performance. It would be interesting to see if the performance can keep going up by feeding more and more human data, or is there somewhere it saturates.

**Questions:**

1. In Figure 4, VPDD w/o human videos is still better than other baselines. What's the reason here? Assuming VPDD-w/o-human has seen the same robot data as other baselines, what's the main factor that causes such a difference?

2. The authors tried VPDD on unseen scenes such as varying background and camera poses. What about unseen tasks? Assuming there are some tasks presented in human videos but not in robot videos, will the model learn to do these tasks, at least to some extent?

3. As a bonus, it would be good to evaluate some off-the-shelf pre-trained video generation models such as Stable Video Diffusion for future exploration, but I don't think it affects the main conclusions of this paper.

**Limitations:**

See above.

---

> ### Author Rebuttal · Authors · 2024-08-07
>
> We thank the reviewer for the thorough review and a positive assessment of our work! We carefully address your concerns as follows.
>
> >1. ... how many robot videos are we using here? If the robot videos are too few, will it affect the resulting pre-trained model from this stage?...
>
> As stated in Section 3.1 of the paper, we use only a few robot demonstrations compared to large-scale human videos to demonstrate the sample efficiency of our method. For Metaworld, we have approximately $2e^5$ frames (50 tasks$\times$ 20 episodes$\times$ 200 steps), which are only 5% of the human videos; For RLBench, we have around $1e^5$ frames, accounting for 2.5% of the human videos. Although the action-labeled data is very limited, our method achieves the best performance compared with the extensive baselines.
>
> We would like to clarify that the main motivation of our paper is to make the best use of the actionless video data for downstream decision making, rather than assuming the robot videos are inaccessible. However, it is reasonable to expect that using less robot data in the pre-training stage will result in worse model performance. This is because the representation $z_{\tilde{{x}}_{0}^{\rm v}}$ provided for finetuning would be less accurate and contain noise that affects the performance. To validate the performance of VPDD without seeing robot data during pre-training, we add an experiment where no robot data is used in the pre-training stage, and evaluate its performance on five tasks in Metaworld. **See the results in Table 1 of the global pdf**. We observe that while the performance of VPDD-w/o.-robot decreases, it remains competitive compared to the baselines.
>
>
>
> >2. ... it would be good to see how the performance changes when trained on different amounts of human data ...
>
> Thanks for your constructive suggestions! We add additional experiments following your advice. Given limited time and resources, we re-train our model using 0%, 1/3 and 2/3 of the human data. We find that while the multi-task decision-making performances are not significantly different, the generalization abilities distinctly improve with more human data. **The added experimental results are refered to in Figure 1 of the global pdf**.
>
> >3. In Figure 4, VPDD w/o human videos is still better than other baselines. What's the reason here? Assuming VPDD-w/o-human has seen the same robot data as other baselines, what's the main factor that causes such a difference?
>
> Thanks for the question. We would like to clarify that, without human videos, VPDD still includes three-stage training (i.e., video tokenization, video prediction, and policy fine-tuning) but is trained solely on robot data. As a result, VPDD-w/o.-human also learns to predict future videos in the robot domain and thus capture significant features for subsequent finetuning. Indeed, the video prediction becomes less generalizable due to the limited video data, while it still provides some foresight through future predictions. Meanwhile, VPDD adopts advanced discrete diffusion and carefully designed transformer architectures with action tokenization for action sequence prediction, which has proved to be more efficient than continuous diffusion in high-dimensional trajectory modeling [1-3].
>
> **Reference**
>
> [1] Vector quantized diffusion model for text-to-image synthesis. CVPR. 2022
>
> [2] Diffsound: Discrete diffusion model for text-to-sound generation. IEEE/ACM Transactions on Audio, Speech, and Language Processing, 2023
>
> [3] Diffusion models: A comprehensive survey of methods and applications[J]. ACM Computing Surveys, 2023
>
> >4. The authors tried VPDD on unseen scenes such as varying background and camera poses. What about unseen tasks? Assuming there are some tasks presented in human videos but not in robot videos, will the model learn to do these tasks, at least to some extent?
>
> Thank you for the question. There are two cases to analyze the generalization to unseen tasks: (a) Tasks that are unseen in pre-training but seen in fine-tuning. The experiments in Q1 sufficiently demonstrate that VPDD can generalize to such unseen tasks since the model has learned similar behaviors from human videos. To further validate the generalizability of VPDD in this case, we consider a realistic setting where only incomplete robot tasks are seen during pre-training. During the pre-training stage, we train the VPDD on 47 tasks on MetaWorld and leave 3 unseen tasks to test the generalizability. In finetuning stage, we fine-tune the pretrained model on these 3 tasks. We report the success rate of the final model trained over 0.2M gradient steps. We can see the performance gap compared with the oracle (tasks seen in pre-training) is small.
>
> | Unseen Tasks   | VPDD | VPDD (oracle) |
> | -------------- | ---- | --- |
> | hand-insert-v2 |   32%   |  36%   |
> | bin-picking-v2 |  78%    |  84%   |
> | door-unlock-v2 |  100%    | 100%    |
>
> (b) When tasks are unseen in both pre-training and fine-tuning, the model struggles to choose the correct actions. It is challenging because it does not know the action space and has no information about the meaning of specific actions. As a result, VPDD struggles to achieve high success rates in such scenarios.
> >5. ... it would be good to evaluate some off-the-shelf pre-trained video generation models such as Stable Video Diffusion for future exploration...
>
> Thank you for your insightful suggestions! While it is a promising idea, it requires extensive revisions that are impractical within the rebuttal period. One potential approach is to add an action-prediction head to Stable Video Diffusion (SVD) and then apply our pre-training and fine-tuning methods. However, multi-modal training with both actions and videos presents significant challenges for SVD and other models, so we will leave this as future work.
>
> We hope to have addressed all the raised concerns. Thank you again for your time and valuable feedback!

---

> > ### Comment · Reviewer_v4Bj · 2024-08-11
> > **Thank you for the response**
> >
> > I would like to thank the authors for their thoughtful response. My concerns are all addressed and I increase my score to 7.

---

> > > ### Author Response · Authors · 2024-08-11
> > > **Thank you for rasing your score!**
> > >
> > > Thank you for increasing your score! We sincerely appreciate your thoughtful and detailed feedback for improving the quality of our paper. We are grateful for your recognition of our efforts and honored by your positive evaluation!

---

### Author Rebuttal · Authors · 2024-08-07

## General Response
We thank all of the reviewers for their time and insightful comments. Furthermore, we are very glad to find that reviewers generally recognized our interesting idea, the superior performance of our method, and the clear presentation of our paper:
### Contributions:
- **Method**: The paper is well-motivated [**v4Bj,kd9D**]. The paper addresses a very timely problem [**ew7s**]. The proposed idea is very interesting [**wSvy**]. The approach is easy for researchers to implement and reproduce results [**ew7s**]. The design of the model and the training pipeline is sensible [**v4Bj**].
- **Presentation**: The paper is well written, clear and easy to understand [**kd9D**]. Clear explanation of the methods [**wSvy**].
- **Experiments**: The paper achieves strong performance on a number of manipulation benchmarks outperforming many baselines [**wSvy**]. Quantitative results for both Meta-World and RLBench are strong over baselines [**ew7s,kd9D**]. The results in are quite promising [**ew7s,v4Bj**]. The experiment results are statistically significant with multiple runs reported accompanied by error bars [**kd9D**].

We thank all the reviewers for their helpful and constructive feedback to improve the quality of our work. We will carefully update our paper in the next version to incorporate the valuable suggestions of the reviewers. In summary, we add the following new experiments to address the reviewers' concerns:

- [**v4Bj, kd9D, wSvy**] We report the performance of VPDD without seeing robot videos in the pretraining stage (see Table 1 of the global pdf).
- [**v4Bj, wSvy**] We ablate the number of human videos used during pre-training (see Figure 1 of the global pdf).
- [**v4Bj**] The generalizability of VPDD for unseen tasks.
- [**kd9D, ew7s**] New experimental results on language table dataset compared with Octo-Base model.
- [**wSvy**] Video prediction performance on Ego4d domain.
- [**wSvy**] We report the performance of an added baseline of the VideoMAE-based diffusion model.

We hope to have addressed all the raised concerns and would be happy to respond to further questions and suggestions.

---

### Decision · Program_Chairs · 2024-09-25

**Decision:**

Accept (poster)

**Comment:**

The paper proposes an approach to leverage action-less human videos to train robot policies. The paper received unanimous acceptance recommendation from the reviewers. The paper is very well motivated and presents a relatively simple approach with strong performance compared to the baselines. The AC follows the recommendation of the reviewers and recommends acceptance. However, reporting results beyond table-top manipulation would make the paper stronger. The authors are encouraged to incorporate the comments of the reviewers and revise the paper.